# $\omega$GNNs: Deep Graph Neural Networks Enhanced by Multiple Propagation Operators

## Abstract

Graph Neural Networks (GNNs) are limited in their propagation operators. These operators often contain non-negative elements only and are shared across channels and layers, limiting the expressiveness of GNNs. Moreover, some GNNs suffer from over-smoothing, limiting their depth. On the other hand, Convolutional Neural Networks (CNNs) can learn diverse propagation filters, and phenomena like over-smoothing are typically not apparent in CNNs. In this paper, we bridge this gap by incorporating trainable channel-wise weighting factors $\omega$ to learn and mix multiple smoothing and sharpening propagation operators at each layer. Our generic method is called $\omega$GNN, and we study two variants: $\omega$GCN and $\omega$GAT. For $\omega$GCN, we theoretically analyse its behaviour and the impact of $\omega$ on the obtained node features. Our experiments confirm these findings, demonstrating and explaining how both variants do not over-smooth. Additionally, we experiment with 15 real-world datasets on node- and graph-classification tasks, where our $\omega$GCN and $\omega$GAT perform better or on par with state-of-the-art methods.

## 1 Introduction

Graph Neural Networks (GNNs) are useful for a wide array of fields, from computer vision and graphics (Monti et al., 2017; Wang et al., 2018; Eliasof & Treister, 2020) and social network analysis (Kipf & Welling, 2016; Defferrard et al., 2016) to bio-informatics (Hamilton et al., 2017; Jumper et al., 2021). Most GNNs are defined by applications of propagation and point-wise operators, where the former is often fixed and based on the graph Laplacian (e.g., GCN (Kipf & Welling, 2016)), or is defined by an attention mechanism (Veličković et al., 2018; Kim & Oh, 2021; Brody et al., 2022).

Most recent GNNs follow a general structure that involves two main ingredients – the propagation operator, denoted by $\mathbf{S}^{(l)}$, and a $1 \times 1$ convolution denoted by $\mathbf{K}^{(l)}$, as follows

$$\mathbf{f}^{(l+1)} = \sigma(\mathbf{S}^{(l)}\mathbf{f}^{(l)}\mathbf{K}^{(l)}), \tag{1}$$

where $\mathbf{f}^{(l)}$ denotes the feature tensor at the $l$-th layer. The main limitation of the above formulation is that the propagation operators in most common architectures are constrained to be non-negative. This leads to two drawbacks. First, this limits the expressiveness of GNNs. For example, the gradient of given graph node features can not be expressed by a non-negative operator, while a mixed-sign operator as in our proposed method can (see demonstrations in Fig. 1 and Fig. 2). Moreover, the utilization of strictly non-negative propagation operators yields a smoothing process, that may lead GNNs to suffer from over-smoothing. That is, the phenomenon where node features become indistinguishable from one and other as more GNN layers are stacked – causing severe performance degradation in deep GNNs (Li et al., 2018; Wu et al., 2019; Wang et al., 2019).

Both of the drawbacks mentioned above are not evident in Convolutional Neural Networks (CNNs), which can be interpreted as structured versions of GNNs (i.e., GNNs operating on a regular grid). The structured convolutions in CNNs allow to learn diverse propagation operators, and in particular it is known that mixed-sign kernels like sharpening filters are useful feature extractors in CNNs (Krizhevsky et al., 2012), and such operators cannot be obtained by non-negative (smoothing) kernels only. In the context of GNNs, Eliasof et al. (2022) have shown the significance and benefit of employing mixed-sign propagation operators in GNNs as well. In addition, the over-smoothing phenomenon is typically not evident in standard CNNs where the propagation (spatial) filters are

**Figure 1:** The impulse response of $\omega$GCN's propagation operator for different $\omega$ values. For $\omega = 0.5, 1.0$ non-negative values are obtained, while for $\omega = 1.5$ we see mixed-sign values. The dashed node starts from a feature of 1 and the rest with 0.

**(a)** Input graph. **(b)** Node gradient. **(c)** GCN estimation. **(d)** $\omega$GCN estimation.

**Figure 2:** The expressiveness of $\omega$GNNs. Our $\omega$GCN can express the gradient of the node features while GCN cannot.

learnt, and usually adding more layers improves accuracy (He et al., 2016). The discussion above demonstrates two gaps between CNNs and GNNs that we seek to bridge in this work.

A third gap between GNNs and CNNs is the ability of the latter to learn and mix multiple propagation operators. In the scope of separable convolutions, CNNs typically learn a distinct kernel per channel, known as a depth-wise convolution (Sandler et al., 2018) – a key element in modern CNNs (Tan & Le, 2019; Liu et al., 2022). On the contrary, the propagation operator $\mathbf{S}^{(l)}$ from equation 1 acts on all channels (Chen et al., 2020b; Veličković et al., 2018), and in some cases on all layers (Kipf & Welling, 2016; Wu et al., 2019). We note that one exception is the multi-head GAT (Veličković et al., 2018) where several attention heads are learnt per layer. However, this approach typically employs only a few heads due to the high computational cost and is still limited by learning non-negative propagation operators only.

In this paper we propose an effective modification to GNNs to directly address the three shortcomings of GNNs discussed above, by introducing a parameter $\omega$ to control the contribution and type of the propagation operator. We call our general approach $\omega$GNN, and utilize GCN (Kipf & Welling, 2016) and GAT (Veličković et al., 2018) to construct two variants, $\omega$GCN and $\omega$GAT. We theoretically prove and empirically demonstrate that our $\omega$GNN can prevent over-smoothing. Secondly, we show that by learning $\omega$, our $\omega$GNNs can yield propagation operators with mixed signs, ranging from smoothing to sharpening operators which do not exist in current GNNs (see Fig. 1 for an illustration). This approach enhances the expressiveness of the network, as demonstrated in Fig. 2, and to the best of our knowledge, was not considered in the GNNs mentioned above that employ non-negative propagation operators only. Lastly, we propose and demonstrate that by learning different $\omega$ per layer and channel, similarly to a depth-wise convolution in CNNs, our $\omega$GNNs obtains state-of-the-art accuracy.

Our contributions are summarized as follows:

- We propose $\omega$GNN, an effective and computationally light modification to GNNs of a common and generic structure, that directly avoids over-smoothing and enhances the expressiveness of GNNs. Our method is demonstrated by $\omega$GCN and $\omega$GAT.
- A theoretical analysis and experimental validation of the behaviour of $\omega$GNN are provided to expose its improved expressiveness compared to standard propagation operators in GNNs.
- We propose to learn multiple propagation operators by learning $\omega$ *per layer and per channel* and mixing them using a $1 \times 1$ convolution to enhance the performance of GNNs.
- Our experiments with 15 real-world datasets on numerous applications and settings, from semi- and fully-supervised node classification to graph classification show that our $\omega$GCN and $\omega$GAT read on par or better performance than current state-of-the-art methods.

## 2 METHOD

We start by providing the notations that will be used throughout this paper, and displaying our general $\omega$GNN in Sec. 2.1. Then we consider two popular GNNs that adhere to the structure presented in equation 1, namely GCN and GAT. We formulate and analyse the behaviour of their two counterparts $\omega$GCN and $\omega$GAT in Sec. 2.2 and 2.3, respectively.

**Notations.** Assume we are given an undirected graph defined by the set $\mathcal{G} = (\mathcal{V}, \mathcal{E})$ where $\mathcal{V}$ is a set of $n$ vertices and $\mathcal{E}$ is a set of $m$ edges. Let us denote by $\mathbf{f}_i \in \mathbb{R}^c$ the feature vector of the $i$-th node of $\mathcal{G}$ with $c$ channels. Also, we denote the adjacency matrix $\mathbf{A}$, where $\mathbf{A}_{ij} = 1$ if there exists an edge $(i,j) \in \mathcal{E}$ and 0 otherwise. We also define the diagonal degree matrix $\mathbf{D}$ where $\mathbf{D}_{ii}$ is the degree of the $i$-th node. The graph Laplacian is given by $\mathbf{L} = \mathbf{D} - \mathbf{A}$. Let us also denote the adjacency and degree matrices with added self-loops by $\tilde{\mathbf{A}}$ and $\tilde{\mathbf{D}}$, respectively. Lastly, we denote the symmetrically normalized graph Laplacian by $\tilde{\mathbf{L}}^{sym} = \tilde{\mathbf{D}}^{-\frac{1}{2}} \tilde{\mathbf{L}} \tilde{\mathbf{D}}^{-\frac{1}{2}}$ where $\tilde{\mathbf{L}} = \tilde{\mathbf{D}} - \tilde{\mathbf{A}}$.

## 2.1 $\omega$GNNs

The goal of $\omega$GNNs is to utilize learnable mixed-sign propagation operators that control smoothing and sharpening to enrich GNNs expressiveness. Below, we describe how the learnt $\omega$ influences the obtained operator and how to learn and mix multiple operators for enhanced expressiveness.

**Learning propagation weight $\omega$.** To address the expressiveness and over-smoothing issues, we suggest a general form given an arbitrary non-negative and normalized (e.g., such that its row sums equal to 1) propagation operator $\mathbf{S}^{(l)}$. Our general $\omega$GNN is then given by

$$\mathbf{f}^{(l+1)} = \sigma\left(\left(\mathbf{I} - \omega^{(l)}\left(\mathbf{I} - \mathbf{S}^{(l)}\right)\right)\mathbf{f}^{(l)}\mathbf{K}^{(l)}\right), \tag{2}$$

where $\omega^{(l)}$ is a scalar that is learnt per layer, and in the next paragraph we offer a more elaborated version with a parameter $\omega$ per layer and channel. The introduction of $\omega^{(l)}$ allows our $\omega$GNN layer to behave in a three-fold manner. When $\omega^{(l)} \leq 1$, a smoothing process is obtained [1]. Note, that for $\omega^{(l)} = 1$, equation 2 reduces to the standard GNN dynamics from equation 1. In case $\omega^{(l)} = 0$, equation 2 reduces to a $1 \times 1$ convolution followed by a non-linear activation function, and does not propagate neighbouring node features. On the other hand, if $\omega^{(l)} > 1$, we obtain an operator with negative signs on the diagonal but positive on the off-diagonal entries, inducing a sharpening operator. An example of various $\omega^{(l)}$ values and their impulse response is given in Fig. 1. Thus, a learnable $\omega^{(l)}$ allows to learn a new family of operators, namely sharpening operators, that are not achieved by methods like GCN and GAT. To demonstrate the importance of sharpening operators, we consider a synthetic task of node gradient feature regression, given a graph and input node features(see Appendix B for more details). As depicted in Fig. 2, using a non-negative operator as in GCN cannot accurately express the gradient operator output, while our $\omega$GCN estimates the gradient output with a machine precision accuracy. Also, the benefit of employing both smoothing and sharpening operators is reflected in the obtained accuracy of our method on real-world datasets in Sec. 4.

**Multiple propagation operators.** To learn multiple propagation operators, we extend equation 2 from a channels-shared weight to channel-wise weights by learning a vector $\vec{\omega}^{(l)} \in c$ as follows

$$\mathbf{f}^{(l+1)} = \sigma\left(\left(\mathbf{I} - \mathbf{\Omega}_{\vec{\omega}^{(l)}}\left(\mathbf{I} - \mathbf{S}^{(l)}\right)\right)\mathbf{f}^{(l)}\mathbf{K}^{(l)}\right), \tag{3}$$

where $\mathbf{\Omega}_{\vec{\omega}^{(l)}}$ is an operator that scales each channel $j$ with a different $\omega_j^{(l)}$. As discussed in Sec. 1, this procedure yields a propagation operator per-channel, which is similar to depth-wise convolutions in CNNs (Howard et al., 2017; Sandler et al., 2018). Thus, the extension to a vector $\vec{\omega}^{(l)}$ helps to further bridge the gap between GNNs and CNNs.

We note that using this approach, our $\omega$GNN is suitable to many existing GNNs, and in particular to those which act as a separable convolution, as described in equation 1. In what follows, we present and analyse two variants based on GCN and GAT, called $\omega$GCN and $\omega$GAT, respectively.

## 2.2 $\omega$GCN

GCNs are a class of GNNs that employ a pre-determined propagation operator $\tilde{\mathbf{P}} = \tilde{\mathbf{D}}^{-\frac{1}{2}} \tilde{\mathbf{A}} \tilde{\mathbf{D}}^{-\frac{1}{2}}$, that stems from the graph Laplacian. For instance, GCN Kipf & Welling (2016) is given by:

$$\mathbf{f}^{(l+1)} = \sigma(\tilde{\mathbf{P}}\mathbf{f}^{(l)}\mathbf{K}^{(l)}), \tag{4}$$

---

[1]The use of the value 1 in this discussion corresponds to a non-negative operator $\mathbf{S}^{(l)}$ with zeros on its diagonal, normalized to have row sums of 1. Other normalizations may yield other constants. Also, if $0 < \mathbf{S}_{ii}^{(l)} < 1$, then setting $\omega^{(l)} > \frac{1}{1-\mathbf{S}_{ii}^{(l)}}$ flips the sign of the $i$-th diagonal entry.

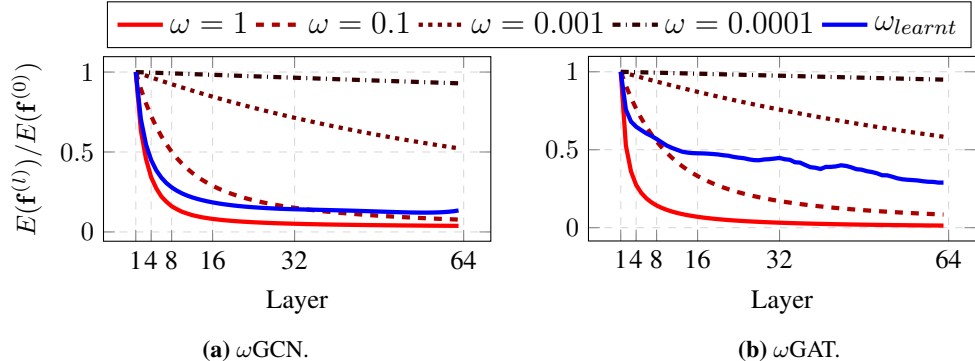

**Figure 3:** Node features energy at the $l$-th layer relative to the initial node embedding energy on Cora. Both $\omega$GCN and $\omega$GAT control the respective energies from equation 6 and equation 11 to avoid over-smoothing, while the baselines with $\omega = 1$ reduce the energies to 0 and over-smooth.

that is, by setting $\mathbf{S}^{(l)} = \tilde{\mathbf{P}}$ in equation 1. Other methods like SGC (Wu et al., 2019), GCNII (Chen et al., 2020b) and EGNN (Zhou et al., 2021) also rely on $\tilde{\mathbf{P}}$ as a propagation operator.

The operator $\tilde{\mathbf{P}}$ is a fixed non-negative smoothing operator, hence, repeated applications of equation 4 lead to the over-smoothing phenomenon, where the feature maps converge to a single eigenvector as shown by Wu et al. (2019); Wang et al. (2019). Moreover, $\tilde{\mathbf{P}}$ is pre-determined, and solely depends on the graph connectivity, disregarding the node features, which may harm performance.

By baking our proposed $\omega$GNN with a learnable weight, denoted by $\omega^{(l)} \in \mathbb{R}$ into GCN we obtain the following propagation scheme, named $\omega$GCN:

$$\mathbf{f}^{(l+1)} = \sigma\left(\left(\mathbf{I} - \omega^{(l)}\left(\mathbf{I} - \tilde{\mathbf{P}}\right)\right)\mathbf{f}^{(l)}\mathbf{K}^{(l)}\right). \tag{5}$$

We now present theoretical analyses of our $\omega$GCN and reason about its *non* over-smoothing property. We first define the node features Dirichlet energy at the $l$-th layer, as in Zhou et al. (2021):

$$E(\mathbf{f}^{(l)}) = \sum_{i \in \mathcal{V}} \sum_{j \in \mathcal{N}_i} \frac{1}{2} \left\| \frac{\mathbf{f}_i^{(l)}}{\sqrt{(1+d_i)}} - \frac{\mathbf{f}_j^{(l)}}{\sqrt{(1+d_j)}} \right\|_2^2. \tag{6}$$

Fig. 3a demonstrates how the Dirichlet energy $E(\mathbf{f}^{(l)})$ decays to zero when $\omega$ is a constant, and to a fixed positive value when $\omega$ is learnt. Next, we provide a theorem that characterizes the behaviour of $\omega$ and how it prevents over-smoothing. To this end we denote the propagation operator of $\omega$GCN from equation 5 by

$$\tilde{\mathbf{P}}_\omega = \mathbf{I} - \omega\left(\mathbf{I} - \tilde{\mathbf{P}}\right) = \mathbf{I} - \omega\left(\mathbf{I} - \tilde{\mathbf{D}}^{-\frac{1}{2}}\tilde{\mathbf{A}}\tilde{\mathbf{D}}^{-\frac{1}{2}}\right) = \mathbf{I} - \omega\tilde{\mathbf{D}}^{-\frac{1}{2}}\mathbf{L}\tilde{\mathbf{D}}^{-\frac{1}{2}}, \tag{7}$$

where the latter equality is shown in Appendix A. In essence, we show that repeatedly applying the operator $\tilde{\mathbf{P}}$ is equivalent to applying gradient descent steps for minimizing equation 6 with a learning rate $\omega$. We build on the observation that smoothing is beneficial (Gasteiger et al., 2019; Chamberlain et al., 2021) and assume that there exists an optimal energy at the last layer that satisfies $0 < E_{opt}(\mathbf{f}^{(L)}) < E(\mathbf{f}^{(0)})$. Then, we show that if we learn a single $\omega^{(l)} = \omega > 0$, shared across all layers, then taking $L$ to infinity will lead $\omega$ to zero. Thus, our $\omega$GCN will not over-smooth, as the energy at the last layer $E(\mathbf{f}^{(L)})$ can reach to $E_{opt}(\mathbf{f}^{(L)})$. Later, in Corollary 1.1, we generalize this result for a per-layer $\omega^{(l)}$, and empirically validate both results in Sec. 4.5. The proofs for the Theorem and Corollary below are given in Appendix A.

**Theorem 1.** *Consider $L$ applications of equation 7, i.e., $\mathbf{f}^{(L)} = (\tilde{\mathbf{P}}_\omega)^L \mathbf{f}^{(0)}$ with a shared parameter $\omega^{(l)} = \omega$ that is used in all the layers. Also assume that there is some optimal Dirichlet energy of the final feature map that satisfies $0 < E_{opt}(\mathbf{f}^{(L)}) < E(\mathbf{f}^{(0)})$. Then, at the limit, as more layers are added, $\omega$ converges to $\bar{\omega}/L$ up to first order accuracy, where $L$ is the number of layers and $\bar{\omega}$ is a value that is independent of $L$ and leads to $E_{opt}$.*

**Corollary 1.1.** *Allowing a variable $\omega^{(l)} > 0$ at each layer in Theorem 1, yields $\sum_{l=0}^{L-1} \omega^{(l)} = \bar{\omega}$ up to first order accuracy.*

Next, we dwell on the second mechanism in which $\omega$GCN prevents over-smoothing. We analyse the eigenvectors of $\tilde{\mathbf{P}}_\omega$, showing that different choices of $\omega$ yield different leading eigenvectors that alter the behaviour of the propagation operator (i.e. smoothing and sharpening processes). This result is useful because changing the leading eigenvector prevents the gravitation towards a specific eigenvector, which causes the over-smoothing to occur (Wu et al., 2019; Oono & Suzuki, 2020).

**Theorem 2.** *Assume that the graph is connected. Then, there exists some $\omega_0 \geq 1$ where for all $0 < \omega < \omega_0$, the operator $\tilde{\mathbf{P}}_\omega$ in equation 7 is smoothing and the leading eigenvector is $\tilde{\mathbf{D}}^{\frac{1}{2}}\mathbf{1}$. For $\omega > \omega_0$ or $\omega < 0$, the leading eigenvector changes.*

The proof for the theorem is given in Appendix A.

**$\omega$GCN with multiple propagation operators.** To further increase the expressiveness of our $\omega$GCN we extend $\omega^{(l)} \in \mathbb{R}$ to $\vec{\omega}^{(l)} \in \mathbb{R}^c$ and learn a propagation operator per channel, at each layer. To this end, we modify equation 5 to the following formulation

$$\mathbf{f}^{(l+1)} = \sigma\left(\left(\mathbf{I} - \mathbf{\Omega}_{\vec{\omega}^{(l)}}\left(\mathbf{I} - \tilde{\mathbf{P}}\right)\right)\mathbf{f}^{(l)}\mathbf{K}^{(l)}\right). \tag{8}$$

As we show in Sec. 4.5, learning a propagation operator per channel is beneficial to improve accuracy.

## 2.3 $\omega$GAT

The seminal GAT (Veličković et al., 2018) learns a non-negative edge-weight as follows

$$\alpha_{ij}^{(l)} = \frac{\exp\left(\text{LeakyReLU}\left(\mathbf{a}^{(l)\top}[\mathbf{W}^{(l)}\mathbf{f}_i^{(l)}||\mathbf{W}^{(l)}\mathbf{f}_j^{(l)}]\right)\right)}{\sum_{p \in \mathcal{N}_i} \exp\left(\text{LeakyReLU}\left(\mathbf{a}^{(l)\top}[\mathbf{W}^{(l)}\mathbf{f}_i^{(l)}||\mathbf{W}^{(l)}\mathbf{f}_p^{(l)}]\right)\right)}, \tag{9}$$

where $\mathbf{a}^{(l)} \in \mathbb{R}^{2c}$ and $\mathbf{W}^{(l)} \in \mathbb{R}^{c \times c}$ are trainable parameters and $||$ denotes channel-wise concatenation. Here, GAT is obtained by defining the propagation operator $\mathbf{S}^{(l)}$ in equation 1 as $\hat{\mathbf{S}}_{ij}^{(l)} = \alpha_{ij}$.

To avoid repeated equations, we skip the per-layer $\omega$ formulation (as in equation 2) and directly define the per-channel $\omega$GAT as follows

$$\mathbf{f}^{(l+1)} = \sigma\left(\left(\mathbf{I} - \mathbf{\Omega}_{\vec{\omega}^{(l)}}\left(\mathbf{I} - \hat{\mathbf{S}}^{(l)}\right)\right)\mathbf{f}^{(l)}\mathbf{K}^{(l)}\right). \tag{10}$$

The introduction of $\mathbf{\Omega}_{\vec{\omega}^{(l)}}$ yields a learnable propagation operator per layer and channel. We note that it is also possible to obtain multiple propagation operators from GAT by using a multi-head attention. However, we distinguish our proposition from GAT in a 2-fold fashion. First, our propagation operators belong to a broader family that includes smoothing and sharpening operators as opposed to smoothing-only due to the SoftMax normalization in GAT. Secondly, our method requires less computational overhead when adding more propagation operators, as our $\omega$GAT requires a scalar per operator, while GAT doubles the number of channels to obtain more attention-heads. Also, utilizing a multi-head GAT can still lead to over-smoothing, as all the heads induce a non-negative operator.

To study the behaviour of our $\omega$GAT, we inspect its node features energy compared to GAT. To this end, we define the GAT energy as

$$E_{\text{GAT}}(\mathbf{f}^{(l)}) = \sum_{i \in \mathcal{V}} \sum_{j \in \mathcal{N}_i} \frac{1}{2}||\mathbf{f}_i^{(l)} - \mathbf{f}_j^{(l)}||_2^2. \tag{11}$$

This modification of the Dirichlet energy from equation 6 is required because in GAT (Veličković et al., 2018) the leading eigenvector of the propagation operator $\hat{\mathbf{S}}^{(l)}$ is the constant vector $\mathbf{1}$ as shown by Chen et al. (2020a), unlike the vector $\tilde{\mathbf{D}}^{\frac{1}{2}}\mathbf{1}$ in the symmetric normalized $\tilde{\mathbf{P}}$ from GCN (Kipf & Welling, 2016) where the Dirichlet energy is natural to consider (Pei et al., 2020).

We present the energy of a 64 layer GAT trained on the Cora dataset in Fig. 3b. It is evident that the accuracy degradation of a deep GAT reported by Zhao & Akoglu (2020) is in congruence with the decaying energy in equation 11, while our $\omega$GAT does not experience decaying energy nor accuray degradation as more layers are added, as can be seen in Tab. 2. To further validate our findings, we repeat this experiment in Appendix D on additional datasets and reach to the same conclusion.

### 2.4 COMPUTATIONAL COSTS

Our $\omega$GNN approach is general and can be applied to any GNN that conforms to the structure of equation 1 and can be modified into equation 3. The additional parameters compared to the baseline GNN are the added $\mathbf{\Omega}_{\vec{\sigma}(l)} \in \mathbb{R}^c$ parameters at each layer, yielding a relatively low computational overhead. For example, in GCN (Kipf & Welling, 2016) there are $c \times c$ trainable parameters requiring $c \times c \times n$ multiplications due to the $1 \times 1$ convolution $\mathbf{K}^{(l)}$. In our $\omega$GCN, we will have $c \times c + c$ parameters and $(c + 1) \times c \times n$ multiplications. That is in addition to applying the propagation operators $\mathbf{S}^{(l)}$, which are identical for both methods. A similar analysis holds for GAT. To validate the actual complexity of our method, we present the training and inference times for $\omega$GCN and $\omega$GAT in Appendix G. We see a negligible addition to the runtimes compared to the baselines, at the return of better performance.

### 3 OTHER RELATED WORK

**Over-smoothing in GNNs.** The over-smoothing phenomenon was identified by Li et al. (2018), and was profoundly studied in recent years. Various methods stemming from different approaches were proposed. For example, methods like DropEdge (Rong et al., 2020), PairNorm (Zhao & Akoglu, 2020), and EGNN (Zhou et al., 2021) propose augmentation, normalization and energy-based penalty methods to alleviate over-smoothing, respectively. Other methods like Min et al. (2020) propose to augment GCN with geometric scattering transforms and residual convolutions, and GCNII (Chen et al., 2020b) present a spectral analysis of the smoothing property of GCN (Kipf & Welling, 2016) and propose adding an initial identity residual connection and a decay of the weights of deeper layers, which are also used in EGNN (Zhou et al., 2021).

**Graph Neural Diffusion.** The view of GNNs as a diffusion process has gained popularity in recent years. Methods like APPNP (Klicpera et al., 2019) propose to use a personalized PageRank (Page et al., 1999) algorithm to determine the diffusion of features, and GDC (Gasteiger et al., 2019) imposes constraints on the ChebNet (Defferrard et al., 2016) architecture to obtain diffusion kernels, showing accuracy improvement. Other works like GRAND (Chamberlain et al., 2021), CFD-GCN (Belbute-Peres et al., 2020), PDE-GCN (Eliasof et al., 2021) and GRAND++ (Thorpe et al., 2022) propose to view GNN layers as time steps in the integration process of ODEs and PDEs that arise from a non-linear heat equation, allowing to control the diffusion (smoothing) in the network to prevent over-smoothing. In addition, some GNNs (Eliasof et al., 2021; Rusch et al., 2022) propose a mixture between diffusion and oscillatory processes to avoid over-smoothing by frequency preservation of the features.

**Mixed-sign operators in GNNs.** The importance of mixed-sign operators in GNNs was discussed in Eliasof et al. (2022), where $k$-hop filters and stochastic path sampling mechanisms are utilized. However, such a method requires significantly more computational resources than a standard GNN like Kipf & Welling (2016); Veličković et al. (2018) due to the path sampling strategy and larger filters of 5-hop required for optimal accuracy. However, our $\omega$GNNs perform 1-hop convolutions and as we show in Appendix G, obtain state-of-the-art results without significant added computational costs.

### 4 EXPERIMENTS

We demonstrate our $\omega$GCN and $\omega$GAT on node classification, inductive learning and graph classification tasks. Additionally, we conduct an ablation study of the different configurations of our method and experimentally verify the theorems from Sec. 2. A description of the network architectures is given in Appendix E. We use the Adam (Kingma & Ba, 2014) optimizer in all experiments, and perform grid search to determine the hyper-parameters reported in Appendix F. The objective function in all experiments is the cross-entropy loss, besides inductive learning on PPI (Hamilton et al., 2017) where we use the binary cross-entropy loss. Our code is implemented with PyTorch (Paszke et al., 2019) and PyTorch-Geometric (Fey & Lenssen, 2019) and trained on an Nvidia Titan RTX GPU.

We show that for all the considered tasks and datasets, whose statistics are provided Appendix C, our $\omega$GCN and $\omega$GAT are either better or on par with other state-of-the-art models.

**Table 1:** Summary of semi-supervised node classification accuracy (%)

| Method | GCN | GAT | APPNP | GCNII | GRAND | superGAT | EGNN | $\omega$GCN (Ours) | $\omega$GAT (Ours) |
|---|---|---|---|---|---|---|---|---|---|
| Cora | 81.1 | 83.1 | 83.3 | 85.5 | 84.7 | 84.3 | 85.7 | **85.9** | 84.8 |
| Citeseer | 70.8 | 70.8 | 71.8 | 73.4 | 73.6 | 72.6 | – | 73.3 | **74.0** |
| Pubmed | 79.0 | 78.5 | 80.1 | 80.3 | 71.0 | 81.7 | 80.1 | 81.1 | **81.8** |

**Table 2:** Semi-supervised node classification accuracy (%). – indicates not available results.

| Dataset | Cora | | | | | | Citeseer | | | | | | Pubmed | | | | | |
|---|---|---|---|---|---|---|---|---|---|---|---|---|---|---|---|---|---|---|
| Layers | 2 | 4 | 8 | 16 | 32 | 64 | 2 | 4 | 8 | 16 | 32 | 64 | 2 | 4 | 8 | 16 | 32 | 64 |
| GCN | **81.1** | 80.4 | 69.5 | 64.9 | 60.3 | 28.7 | **70.8** | 67.6 | 30.2 | 18.3 | 25.0 | 20.0 | **79.0** | 76.5 | 61.2 | 40.9 | 22.4 | 35.3 |
| GCN (Drop) | **82.8** | 82.0 | 75.8 | 75.7 | 62.5 | 49.5 | **72.3** | 70.6 | 61.4 | 57.2 | 41.6 | 34.4 | **79.6** | 79.4 | 78.1 | 78.5 | 77.0 | 61.5 |
| JKNet | – | 80.2 | 80.7 | 80.2 | **81.1** | 71.5 | – | 68.7 | 67.7 | **69.8** | 68.2 | 63.4 | – | 78.0 | **78.1** | 72.6 | 72.4 | 74.5 |
| JKNet (Drop) | – | **83.3** | 82.6 | 83.0 | 82.5 | 83.2 | – | 72.6 | 71.8 | **72.6** | 70.8 | 72.2 | – | 78.7 | 78.7 | **79.7** | 79.2 | 78.9 |
| Incep | – | 77.6 | 76.5 | 81.7 | **81.7** | 80.0 | – | 69.3 | 68.4 | **70.2** | 68.0 | 67.5 | – | 77.7 | **77.9** | 74.9 | – | – |
| Incep (Drop) | – | 82.9 | 82.5 | 83.1 | 83.1 | **83.5** | – | **72.7** | 71.4 | 72.5 | 72.6 | 71.0 | – | **79.5** | 78.6 | 79.0 | – | – |
| GCNII | 82.2 | 82.6 | 84.2 | 84.6 | 85.4 | **85.5** | 68.2 | 68.8 | 70.6 | 72.9 | **73.4** | 73.4 | 78.2 | 78.8 | 79.3 | **80.2** | 79.8 | 79.7 |
| GCNII* | 80.2 | 82.3 | 82.8 | 83.5 | 84.9 | **85.3** | 66.1 | 66.7 | 70.6 | 72.0 | **73.2** | 73.1 | 77.7 | 78.2 | 78.8 | **80.3** | 79.8 | 80.1 |
| PDE-GCN$_D$ | 82.0 | 83.6 | 84.0 | 84.2 | 84.3 | **84.3** | 74.6 | 75.0 | 75.2 | 75.5 | **75.6** | 75.5 | 79.3 | **80.6** | 80.1 | 80.4 | 80.2 | 80.3 |
| EGNN | 83.2 | – | – | 85.4 | – | **85.7** | – | – | – | – | – | – | 79.2 | – | – | 80.0 | – | **80.1** |
| $\omega$GCN (Ours) | 82.6 | 83.8 | 84.2 | 84.4 | 85.5 | **85.9** | 71.3 | 71.6 | 72.1 | 72.4 | 73.3 | **73.3** | 79.7 | 80.2 | 80.1 | 80.5 | 80.8 | **81.1** |
| $\omega$GAT (Ours) | 83.4 | 83.7 | 84.0 | 84.3 | 84.4 | **84.8** | 72.5 | 73.1 | 73.3 | 73.5 | 73.9 | **74.0** | 80.3 | 81.0 | 81.2 | 81.3 | 81.5 | **81.8** |

## 4.1 SEMI-SUPERVISED NODE CLASSIFICATION

We employ the Cora, Citeseer and Pubmed (Sen et al., 2008) datasets using the standard training/validation/testing split by Yang et al. (2016), with 20 nodes per class for training, 500 validation nodes and 1,000 testing nodes. We follow the training and evaluation scheme of Chen et al. (2020b) and compare with models like GCN, GAT, superGAT (Kim & Oh, 2021), Inception (Szegedy et al., 2017), APPNP (Klicpera et al., 2019), JKNet Xu et al. (2018), DropEdge (Rong et al., 2020), GCNII (Chen et al., 2020b), GRAND (Chamberlain et al., 2021), PDE-GCN (Eliasof et al., 2021) and EGNN (Zhou et al., 2021). We summarize the results in Tab. 1 where we see better or on par performance with other state-of-the-art methods. Additionally, we report the accuracy per number of layers, from 2 to 64 In Tab. 2, where it is evident that our $\omega$GCN and $\omega$GAT do not over-smooth. To ensure the robustness of our method, we also experiment with 100 random splits in Appendix I where our $\omega$GCN and $\omega$GAT continue to perform better or on par with state-of-the-art methods.

## 4.2 FULLY-SUPERVISED NODE CLASSIFICATION

To further validate the efficacy of our method on fully-supervised node classification, both on homophilic and heterophilic datasets as defined in Pei et al. (2020). Specifically, examine our $\omega$GCN and $\omega$GAT on Cora, Citeseer, Pubmed, Chameleon (Rozemberczki et al., 2021), Cornell, Texas and Wisconsin using the identical train/validation/test splits of $48\%, 32\%, 20\%$, respectively, and report the average performance over 10 random splits from Pei et al. (2020). We compare our performance with, GCN, GAT, Geom-GCN, APPNP, JKNet, Inception, GCNII, PDE-GCN and others, as presented in Tab. 3. Additionally, we evaluate our $\omega$GCN and $\omega$GAT on the Actor (Rozemberczki et al., 2021) and Ogbn-arxiv (Hu et al., 2020) datasets, as reported in Tab. 4. We see an accuracy improvement across all benchmarks compared to the considered methods. In Appendix J we present and discuss the learnt $\vec{\omega}$ for homophilic and heterophilic datasets.

## 4.3 INDUCTIVE LEARNING

We employ the PPI dataset (Hamilton et al., 2017) for the inductive learning task. We use 8 layer $\omega$GCN and $\omega$GAT, with a learning rate of 0.001, dropout of 0.2 and no weight-decay. As a comparison we consider several methods and report the micro-averaged F1 score in in Tab. 5. Our $\omega$GCN achieves

**Table 3:** Fully-supervised node classification accuracy (%). (L) denotes the number of layers.

| Method | Cora | Cite. | Pubm. | Cham. | Corn. | Texas | Wisc. |
|---|---|---|---|---|---|---|---|
| GCN (Kipf & Welling, 2016) | 85.77 | 73.68 | 88.13 | 28.18 | 52.70 | 52.16 | 45.88 |
| GAT (Veličković et al., 2018) | 86.37 | 74.32 | 87.62 | 42.93 | 54.32 | 58.38 | 49.41 |
| Geom-GCN (Pei et al., 2020) | 85.27 | 77.99 | 90.05 | 60.90 | 60.81 | 67.57 | 64.12 |
| APPNP (Klicpera et al., 2019) | 87.87 | 76.53 | 89.40 | 54.30 | 73.51 | 65.41 | 69.02 |
| H2GCN (Zhu et al., 2020) | 87.87 | 77.11 | 89.49 | 60.11 | 82.70 | 84.86 | 87.65 |
| SD (Bodnar et al., 2022) | 87.30 | 74.14 | 89.49 | 68.86 | 86.49 | 85.95 | 89.41 |
| JKNet (Xu et al., 2018) | 85.25 (16) | 75.85 (8) | 88.94 (64) | 60.07 (32) | 57.30 (4) | 56.49 (32) | 48.82 (8) |
| JKNet (Drop) (Rong et al., 2020) | 87.46 (16) | 75.96 (8) | 89.45 (64) | 62.08 (32) | 61.08 (4) | 57.30 (32) | 50.59 (8) |
| Incep (Drop) (Rong et al., 2020) | 86.86 (8) | 76.83 (8) | 89.18 (4) | 61.71 (8) | 61.62 (16) | 57.84 (8) | 50.20 (8) |
| GCNII (Chen et al., 2020b) | 88.49 (64) | 77.08 (64) | 89.57 (64) | 60.61 (8) | 74.86 (16) | 69.46 (32) | 74.12 (16) |
| GCNII* (Chen et al., 2020b) | 88.01 (64) | 77.13 (64) | 90.30 (64) | 62.48 (8) | 76.49 (16) | 77.84 (32) | 81.57 (16) |
| PDE-GCN$_M$ (Eliasof et al., 2021) | 88.60 (16) | **78.48** (32) | 89.93 (16) | 66.01 (16) | 89.73 (64) | 93.24 (32) | 91.76 (16) |
| $\omega$GCN (Ours) | **89.30** (16) | 77.88 (16) | 90.45 (8) | 70.02 (16) | 91.35 (32) | 94.05 (32) | 92.35 (32) |
| $\omega$GAT (Ours) | 89.25 (8) | 78.01 (16) | **90.65** (8) | **72.23** (8) | **91.62** (16) | **94.59** (16) | **92.94** (16) |

**Table 4:** Fully-supervised node classification accuracy (%).

| Method | Actor | Ogbn-arxiv |
|---|---|---|
| GCN (Kipf & Welling, 2016) | 26.86 | 71.74 |
| GAT (Veličković et al., 2018) | 28.45 | 71.59 |
| GATv2 (Brody et al., 2022) | – | 71.87 |
| APPNP (Klicpera et al., 2019) | 31.26 | 71.82 |
| Geom-GCN-P (Pei et al., 2020) | 31.63 | – |
| JKNet (Xu et al., 2018) | 29.81 | 72.19 |
| SGC (Wu et al., 2019) | 30.98 | 69.20 |
| GCNII (Chen et al., 2020b) | 32.87 | 72.74 |
| EGNN (Zhou et al., 2021) | – | 72.70 |
| GRAND (Chamberlain et al., 2021) | – | 72.23 |
| $\omega$GCN (Ours) | **38.94** | **73.02** |
| $\omega$GAT (Ours) | 38.64 | 72.76 |

**Table 5:** Inductive learning on PPI dataset. Results are reported in micro-averaged F1 score.

| Method | Micro-averaged F1 |
|---|---|
| GCN Kipf & Welling (2016) | 60.73 |
| GraphSAGE Hamilton et al. (2017) | 61.20 |
| VR-GCN (Chen et al., 2018) | 97.80 |
| GaAN (Zhang et al., 2018a) | 98.71 |
| GAT (Veličković et al., 2018) | 97.30 |
| JKNet (Xu et al., 2018) | 97.60 |
| GeniePath (Liu et al., 2018) | 98.50 |
| Cluster-GCN (Chiang et al., 2019) | 99.36 |
| GCNII* (Chen et al., 2020b) | 99.58 |
| PDE-GCN$_M$ (Eliasof et al., 2021) | 99.18 |
| $\omega$GCN (Ours) | **99.60** |
| $\omega$GAT (Ours) | 99.48 |

a score of 99.60, which is significantly superior than its baseline – GCN, and also performs better than methods like GAT, JKNet, GeniePath, Cluster-GCN and PDE-GCN.

## 4.4 GRAPH CLASSIFICATION

Previous experiments considered the *node-classification* task. To further demonstrate the efficacy of our $\omega$GNNs we experiment with graph classification on TUDatasets (Morris et al., 2020). Here, we follow the same experimental settings from Xu et al. (2019), and report the 10 fold cross-validation performance on MUTAG, PTC, PROTEINS, NCI1 and NCI109 datasets. The hyper-parameters are determined by a grid search, as in Xu et al. (2019) and are reported in Appendix E. We compare our $\omega$GCN and $\omega$GAT with recent and popular methods like GIN (Xu et al., 2019), DGCNN (Zhang et al., 2018b), IGN (Maron et al., 2018), GSN (Bouritsas et al., 2022), SIN (Bodnar et al., 2021b), CIN (Bodnar et al., 2021a) and others. We also compare with methods that stem from 'classical' graph algorithms like RWK (Gärtner et al., 2003) and WL Kernel (Shervashidze et al., 2011). All the results are summarized in Tab. 6, with an evident improvement or similar results to current deep learning as well as classical methods, highlighting the efficacy of our approach.

## 4.5 ABLATION STUDY

In this section we study the different components and configurations of our $\omega$GNN. We start by allowing a global (single) $\omega$ to be learnt throughout all the layers—this architecture is dubbed as $\omega$GCN$_G$. We validate that this simple variant does not over-smooth, depicted in Tab. 7. The table also shows $\omega$GCN$_{PL}$, that includes a single parameter $\omega^{(l)}$ per layer, and $\omega$GCN shown in the results earlier that has $\Omega^{(l)}$, i.e., a parameter per layer and channel, which yields further accuracy improvements.

**Table 6:** Graph classification accuracy (%) on TUDatasets (Morris et al., 2020).

| Method | MUTAG | PTC | PROTEINS | NCI1 | NCI109 |
|---|---|---|---|---|---|
| RWK (Gärtner et al., 2003) | $79.2 \pm 2.1$ | $55.9 \pm 0.3$ | $59.6 \pm 0.1$ | – | – |
| GK (Shervashidze et al., 2009) | $81.4 \pm 1.7$ | $55.7 \pm 0.5$ | $71.4 \pm 0.3$ | $62.5 \pm 0.3$ | $62.4 \pm 0.3$ |
| PK (Neumann et al., 2016) | $76.0 \pm 2.7$ | $59.5 \pm 2.4$ | $73.7 \pm 0.7$ | $82.5 \pm 0.5$ | – |
| WL Kernel (Shervashidze et al., 2011) | $90.4 \pm 5.7$ | $59.9 \pm 4.3$ | $75.0 \pm 3.1$ | $\mathbf{86.0 \pm 1.8}$ | – |
| DGCNN (Zhang et al., 2018b) | $85.8 \pm 1.8$ | $58.6 \pm 2.5$ | $75.5 \pm 0.9$ | $74.4 \pm 0.5$ | – |
| IGN (Maron et al., 2018) | $83.9 \pm 13.0$ | $58.5 \pm 6.9$ | $76.6 \pm 5.5$ | $74.3 \pm 2.7$ | $72.8 \pm 1.5$ |
| PPGNS (Maron et al., 2019) | $90.6 \pm 8.7$ | $66.2 \pm 6.6$ | $77.2 \pm 4.7$ | $83.2 \pm 1.1$ | $82.2 \pm 1.4$ |
| NATURAL GN (de Haan et al., 2020) | $89.4 \pm 1.6$ | $66.8 \pm 1.7$ | $71.7 \pm 1.0$ | $82.4 \pm 1.3$ | – |
| GSN (Bouritsas et al., 2022) | $92.2 \pm 7.5$ | $68.2 \pm 7.2$ | $76.6 \pm 5.0$ | $83.5 \pm 2.0$ | – |
| SIN (Bodnar et al., 2021b) | – | – | $76.4 \pm 3.3$ | $82.7 \pm 2.1$ | – |
| CIN (Bodnar et al., 2021a) | $92.7 \pm 3.6$ | $68.2 \pm 3.5$ | $77.0 \pm 3.4$ | $83.6 \pm 3.1$ | $84.0 \pm 3.1$ |
| GIN (Xu et al., 2019) | $89.4 \pm 5.6$ | $64.6 \pm 7.0$ | $76.2 \pm 2.8$ | $82.7 \pm 1.7$ | $82.2 \pm 1.6$ |
| GIN + ID (You et al., 2021) | $90.4 \pm 5.4$ | $67.2 \pm 4.3$ | $75.4 \pm 2.7$ | $82.6 \pm 1.6$ | $82.1 \pm 1.5$ |
| DROP (Rong et al., 2020) | $91.0 \pm 5.7$ | $64.5 \pm 2.6$ | $73.5 \pm 4.5$ | $82.0 \pm 2.6$ | $82.2 \pm 1.4$ |
| GCONV (Morris et al., 2019) | $90.5 \pm 4.6$ | $64.9 \pm 10.4$ | $73.9 \pm 6.1$ | $82.4 \pm 2.7$ | $81.7 \pm 1.0$ |
| RNI (Abboud et al., 2020) | $91.0 \pm 4.9$ | $64.3 \pm 6.1$ | $73.3 \pm 3.3$ | $82.1 \pm 1.7$ | $81.7 \pm 1.0$ |
| $\omega$GCN (Ours) | $94.6 \pm 4.1$ | $73.8 \pm 4.3$ | $80.2 \pm 2.5$ | $84.1 \pm 1.2$ | $\mathbf{84.5 \pm 1.8}$ |
| $\omega$GAT (Ours) | $\mathbf{95.2 \pm 3.7}$ | $\mathbf{75.8 \pm 3.5}$ | $\mathbf{80.7 \pm 3.7}$ | $84.4 \pm 1.7$ | $83.6 \pm 1.2$ |

**Table 7:** Accuracy (%) of variants of $\omega$GCN on semi-supervised classification.

| Data. | Variant | Layers | | | | | |
|---|---|---|---|---|---|---|---|
| | | 2 | 4 | 8 | 16 | 32 | 64 |
| Cora | $\omega$GCN$_G$ | 83.4 | 84.3 | 84.2 | 84.1 | 84.3 | 84.4 |
| | $\omega$GCN$_{PL}$ | 83.0 | 83.6 | 84.0 | 84.2 | 84.5 | 84.8 |
| | $\omega$GCN | 82.6 | 83.8 | 84.2 | 84.4 | 85.5 | 85.9 |
| Cite. | $\omega$GCN$_G$ | 71.0 | 71.4 | 71.3 | 71.7 | 72.0 | 71.8 |
| | $\omega$GCN$_{PL}$ | 71.1 | 71.3 | 71.5 | 71.8 | 72.4 | 72.6 |
| | $\omega$GCN | 71.3 | 71.6 | 72.1 | 72.4 | 73.3 | 73.3 |
| Pub. | $\omega$GCN$_G$ | 79.8 | 80.4 | 80.5 | 80.4 | 80.2 | 80.3 |
| | $\omega$GCN$_{PL}$ | 79.8 | 80.0 | 80.2 | 80.4 | 80.5 | 80.8 |
| | $\omega$GCN | 79.7 | 80.2 | 80.1 | 80.5 | 80.8 | 81.1 |

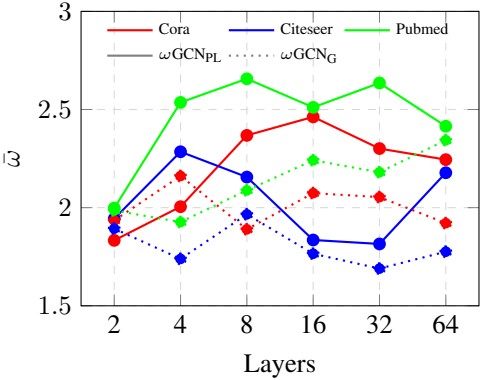

**Figure 4:** The summation of weighting factors $\bar{\omega}$ vs. the number layers for $\omega$GCN$_G$ and $\omega$GCN$_{PL}$.

In addition, we empirically verify our theoretical results from Sec. 2 in Fig. 4, where we show that $\bar{\omega} = L\omega$ and $\bar{\omega} = \sum_{l=0}^{L-1} \omega^{(l)}$ is similar for varying number of layers $L$ as Theorem 1 and Corollary 1.1 suggest. For completeness, we also perform the ablation study on $\omega$GAT in Appendix H.

## 5 SUMMARY

In this work we proposed an effective and computationally efficient modification that applies to a large family of GNNs that carry the form of a separable propagation and $1 \times 1$ convolutions, and in particular we demonstrate its efficacy on the popular GCN and GAT architectures. We provide theorems that reason about the smoothing nature of GCN and through the lens of operator analysis suggest to learn weighting factors $\vec{\omega}$ learn and mix smoothing and sharpening propagation operators. Through an extensive set of experiments on numerous datasets, ranging from node classification to graph classification, as well as an ablation study that validates our theoretical findings, we demonstrate the contribution of our $\omega$GNN, reading on par or achieving new state-of-the-art performance.

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

# A PROOFS OF THEOREMS

Here we repeat the theorems, observations and corollaries from the main paper, for convenience, and provide their proofs or derivation.

$\tilde{\mathbf{P}}$ **is a scaled diffusion operator.** Assume that $\mathbf{A}$ is the adjacency matrix, and $\mathbf{D}$ is the degree matrix. Denote the adjacency matrix with added self-loops by $\tilde{\mathbf{A}} = \mathbf{A} + \mathbf{I}$. Then, the convolution operator from GCN (Kipf & Welling, 2016) is

$$\tilde{\mathbf{P}} = \tilde{\mathbf{D}}^{-\frac{1}{2}} \tilde{\mathbf{A}} \tilde{\mathbf{D}}^{-\frac{1}{2}} \tag{12}$$

We first note that the Laplacian including self loops is the same as the regular Laplacian:

$$\tilde{\mathbf{L}} = \tilde{\mathbf{D}} - \tilde{\mathbf{A}} = \mathbf{D} + \mathbf{I} - \mathbf{A} - \mathbf{I} = \mathbf{D} - \mathbf{A} = \mathbf{L}. \tag{13}$$

Therefore, it holds that:

$$\begin{aligned} \tilde{\mathbf{P}} &= \mathbf{I} - \mathbf{I} + \tilde{\mathbf{D}}^{-\frac{1}{2}} \tilde{\mathbf{A}} \tilde{\mathbf{D}}^{-\frac{1}{2}} \\ &= \mathbf{I} - \tilde{\mathbf{D}}^{-\frac{1}{2}} \tilde{\mathbf{D}} \tilde{\mathbf{D}}^{-\frac{1}{2}} + \tilde{\mathbf{D}}^{-\frac{1}{2}} \tilde{\mathbf{A}} \tilde{\mathbf{D}}^{-\frac{1}{2}} \\ &= \mathbf{I} - \tilde{\mathbf{D}}^{-\frac{1}{2}} (\tilde{\mathbf{D}} - \tilde{\mathbf{A}}) \tilde{\mathbf{D}}^{-\frac{1}{2}} \\ &= \mathbf{I} - \tilde{\mathbf{D}}^{-\frac{1}{2}} (\mathbf{D} - \mathbf{A}) \tilde{\mathbf{D}}^{-\frac{1}{2}} \\ &= \mathbf{I} - \tilde{\mathbf{D}}^{-\frac{1}{2}} \mathbf{L} \tilde{\mathbf{D}}^{-\frac{1}{2}}. \end{aligned} \tag{14}$$

**Theorem 1.** *Consider the L times application of equation 7 from the main paper, i.e., $\mathbf{f}^{(L)} = (\tilde{\mathbf{P}}_\omega)^L \mathbf{f}^{(0)}$ with a shared parameter $\omega^{(l)} = \omega$ that is used in all the layers. Also assume that there is some optimal Dirichlet energy of the final feature map that satisfies $0 < E_{opt}(\mathbf{f}^{(L)}) < E(\mathbf{f}^{(0)})$. Then, at the limit, as more layers are added, $\omega$ converges to $\bar{\omega}/L$ up to first order accuracy, where $L$ is the number of layers and $\bar{\omega}$ is a value that is independent of $L$ and leads to $E_{opt}$.*

*Proof.* First, note that equation 6 from the main paper can be written as

$$E(\mathbf{f}^{(l)}) = \sum_{i \in \mathcal{V}} \sum_{j \in \mathcal{N}_i} \frac{1}{2} \left\| \frac{\mathbf{f}_i^{(l)}}{\sqrt{(1+d_i)}} - \frac{\mathbf{f}_j^{(l)}}{\sqrt{(1+d_j)}} \right\|_2^2 = \frac{1}{2} \| \mathbf{G} \tilde{\mathbf{D}}^{-\frac{1}{2}} \mathbf{f}^{(l)} \|_2^2, \tag{15}$$

where $\mathbf{G}$ is the graph gradient operator, also known as the incidence matrix, that for each edge subtracts the features of the two connected nodes, i.e., $\mathbf{Gf}_{(i,j)}^{(l)} = \mathbf{f}_i^{(l)} - \mathbf{f}_j^{(l)}$ for $(i, j) \in \mathcal{E}$. Let us assume that the initial feature $\mathbf{f}^{(0)}$ has some Dirichlet energy $E_0 > E_{opt}$ as defined in equation 15. Since

$$\nabla E = \tilde{\mathbf{D}}^{-\frac{1}{2}} \mathbf{G}^\top \mathbf{G} \tilde{\mathbf{D}}^{-\frac{1}{2}} \mathbf{f}^{(l)}$$

we see that the forward propagation through a GCN approximates the gradient flow of the Dirichlet energy. That is, for given $L$ and and $\omega$ we have that

$$\mathbf{f}^{(l+1)} = \mathbf{f}^{(l)} - \omega \nabla E = \mathbf{f}^{(l)} - \omega \tilde{\mathbf{D}}^{-\frac{1}{2}} \mathbf{G}^\top \mathbf{G} \tilde{\mathbf{D}}^{-\frac{1}{2}} \mathbf{f}^{(l)} = (\mathbf{I} - \omega \tilde{\mathbf{D}}^{-\frac{1}{2}} \mathbf{L} \tilde{\mathbf{D}}^{-\frac{1}{2}}) \mathbf{f}^{(l)} \tag{16}$$

where we used that $\mathbf{G}^\top \mathbf{G} = \mathbf{L}$. Equation 16 can be seen both as a gradient descent step to reduce $E$, and also as a forward Euler approximation with step size $\omega$ of the solution of

$$\frac{\partial \mathbf{f}(t)}{\partial t} = -\tilde{\mathbf{D}}^{-\frac{1}{2}} \mathbf{L} \tilde{\mathbf{D}}^{-\frac{1}{2}} \mathbf{f}(t), \quad \mathbf{f}(0) = \mathbf{f}^{(0)}. \tag{17}$$

It is known that the solution to equation 17 is given by

$$\mathbf{f}(t) = \exp\left(-t \tilde{\mathbf{D}}^{-\frac{1}{2}} \mathbf{L} \tilde{\mathbf{D}}^{-\frac{1}{2}}\right) \mathbf{f}(0). \tag{18}$$

Since the Dirichlet energy of $\mathbf{f}(t)$ is continuous in $t$ and decays monotonically from $E_0$ to zero, there exists a $T$ such that $E(\mathbf{f}(T)) = E_{opt}$. Now, considering discrete time intervals $0 = t_0, ..., t_L = T$, then, similarly to equation 18, for any two subsequent time steps $t_{l+1}$ and $t_l$ we have that

$$\mathbf{f}(t_{l+1}) = \exp\left(-(t_{l+1} - t_l) \tilde{\mathbf{D}}^{-\frac{1}{2}} \mathbf{L} \tilde{\mathbf{D}}^{-\frac{1}{2}}\right) \mathbf{f}(t_l). \tag{19}$$

Taking fixed-interval time steps such that $t_{l+1} - t_l = \omega = T/L$ for $l = 0, ..., L$, we get

$$\mathbf{f}(t_{l+1}) = \exp\left(-\omega \tilde{\mathbf{D}}^{-\frac{1}{2}} \mathbf{L} \tilde{\mathbf{D}}^{-\frac{1}{2}}\right) \mathbf{f}(t_l) = (\mathbf{I} - \omega \tilde{\mathbf{D}}^{-\frac{1}{2}} \mathbf{L} \tilde{\mathbf{D}}^{-\frac{1}{2}}) \mathbf{f}(t_l) + O(\omega^2), \qquad (20)$$

where the rightmost approximation holds due to to the Taylor expansion up to first order approximation. Denoting $\mathbf{f}^{(l)} = \mathbf{f}(t_l)$ and $\bar{\omega} = T$, we complete the proof. $\qquad\square$

**Corollary 1.** *Allowing a variable $\omega^{(l)} > 0$ in Theorem 1, yields $\sum_{l=0}^{L-1} \omega^{(l)} = \bar{\omega}$ up to first order accuracy.*

*Proof.* The proof follows immediately by setting variable $t_{l+1} - t_l = \omega^{(l)}$ and placing in equation 20. $\qquad\square$

**Remark 1** (The non-negativity of $\tilde{\mathbf{P}}_\omega$). *By definition, for $0 < \omega \le 1$ all the spatial weights of $\tilde{\mathbf{P}}_\omega$ defined in equation 7 are non-negative, and it is that the operator is smoothing as it is a low-pass filter. For $\omega > 1$ or $\omega < 0$, by definition we have an operator with mixed signs.*

**Theorem 2.** *Assume that the graph is connected. Then, there exists some $\omega_0 \ge 1$ where for all $0 < \omega < \omega_0$, the operator $\tilde{\mathbf{P}}_\omega$ in equation 7 from the main paper is smoothing and the leading eigenvector is $\tilde{\mathbf{D}}^{\frac{1}{2}} \mathbf{1}$. For $\omega > \omega_0$ or $\omega < 0$, the leading eigenvector changes.*

*Proof.* Assuming that the graph is connected, it is known that the graph Laplacian matrix has the eigenvector $\mathbf{1}$ whose eigenvalue is 0, i.e. $\mathbf{L1} = 0$. Hence, we get that $\tilde{\mathbf{D}}^{-\frac{1}{2}} \mathbf{L} \tilde{\mathbf{D}}^{-\frac{1}{2}} \tilde{\mathbf{D}}^{\frac{1}{2}} \mathbf{1} = 0$ so $\tilde{\mathbf{D}}^{\frac{1}{2}} \mathbf{1}$ is the eigenvector of the normalized Laplacian with eigenvalue of 0.

Furthermore, denote the normalized Laplacian by $\tilde{\mathbf{L}} = \tilde{\mathbf{D}}^{-\frac{1}{2}} \mathbf{L} \tilde{\mathbf{D}}^{-\frac{1}{2}}$. Consider the range

$$0 < \omega < \frac{2}{\rho(\tilde{\mathbf{L}})} = \omega_0,$$

where $\rho(\tilde{\mathbf{L}})$ denotes the spectral radius of the matrix $\tilde{\mathbf{L}}$. It is easy to verify that for this range of values for $\omega$, the largest eigenvalue in magnitude of $\tilde{\mathbf{P}}_\omega$ is 1, and it corresponds to the null eigenvector of $\tilde{\mathbf{L}}$, i.e., $\tilde{\mathbf{D}}^{\frac{1}{2}} \mathbf{1}$. Hence, for this range, $\tilde{\mathbf{P}}$ is smoothing. For $\omega > \omega_0$ and $\omega < 0$, the leading eigenvector of $\tilde{\mathbf{P}}_\omega$ becomes the leading eigenvector of $\tilde{\mathbf{L}}$. Furthermore, it can be shown that $\rho(\tilde{\mathbf{L}}) \le 2$ (see Williamson (2016) for the proof), hence $\omega_0 \ge 1$. $\qquad\square$

## B   SYNTHETIC EXPRESSIVENESS TASK

To demonstrate the importance and benefit of learning sharpening propagation operators in addition to smoothing operators, we propose the following synthetic node gradient regression task. Given a graph $\mathcal{G} = (\mathcal{V}, \mathcal{E})$ with some input node features $\mathbf{f}^{in} \in \mathbb{R}^{n \times c_{in}}$, we wish a GNN to regress the node features gradient, $\nabla \mathbf{f}^{in}$, where the node feature gradient of the $i$-th node is defined as an upwind gradient operator:

$$\nabla \mathbf{f}_i^{in} = \max_{j \in \mathcal{N}_i} (\mathbf{f}_i - \mathbf{f}_j), \qquad (21)$$

where the goal of the considered GNN is to minimize the following objective:

$$\|\text{GNN}(\mathbf{f}^{in}, \mathcal{G}) - \nabla \mathbf{f}^{in}\|_2^2. \qquad (22)$$

As a comparison, we consider two GNNs: GCN (Kipf & Welling, 2016) and our $\omega$GCN with 64 channels and 2 layers. In both cases we use a learning rate of $1e-4$ without weigh decay and train the network for 5000 iterations (no further benefit was obtained with any of the considered methods). The input graph is a random Erdős–Rényi graph with 8 nodes and an edge rate of $30\%$, with input node features sampled form a uniform distribution in the range of 0 to 1. The obtained loss of GCN is of order $1e-1$, while our $\omega$GCN obtains a loss of order $1e-12$, also as can be seen in Fig. 2. We therefore conclude that introducing the ability of learning mixed-sign operators by $\omega$ is beneficial to enhance the expressiveness of GNNs.

## C  DATASETS

In this section we provide the statistics of the datasets used throughout our experiments. Tab. 8 presents information regarding node-classification datasets, and Tab. 9 summarizes the graph-classification datasets. For each dataset, we also provide the homophily score as defined by Pei et al. (2020).

**Table 8:** Node classification datasets statistics. Hom. score denotes the homophily score.

| Dataset | Cora | Citeseer | Pubmed | Chameleon | Actor (Film) | Cornell | Texas | Wisconsin | PPI | Ogbn-arxiv |
|---|---|---|---|---|---|---|---|---|---|---|
| Classes | 7 | 6 | 3 | 5 | 5 | 5 | 5 | 5 | 121 | 40 |
| Nodes | 2,708 | 3,327 | 19,717 | 2,277 | 7,600 | 183 | 183 | 251 | 56,944 | 169,343 |
| Edges | 5,429 | 4,732 | 44,338 | 36,101 | 33,544 | 295 | 309 | 499 | 818,716 | 1,116,243 |
| Features | 1,433 | 3,703 | 500 | 2,325 | 932 | 1,703 | 1,703 | 1,703 | 50 | 128 |
| Hom. score | 0.81 | 0.80 | 0.74 | 0.23 | 0.22 | 0.30 | 0.11 | 0.21 | 0.17 | 0.63 |

**Table 9:** TUDatasets graph classification statistics.

| Dataset | MUTAG | PTC | PROTEINS | NCI1 | NCI109 |
|---|---|---|---|---|---|
| Classes | 2 | 2 | 2 | 2 | 2 |
| Graphs | 188 | 344 | 1113 | 4110 | 4127 |
| Avg. nodes | 17.93 | 14.29 | 39.06 | 29.87 | 32.13 |
| Avg. edges | 19.79 | 14.69 | 72.82 | 32.30 | 32.13 |

## D  OVER-SMOOTHING IN GAT

In addition to the observation presented in Sec. 2.1 and specifically in Fig. 3b where we see that recurrent applications of GAT reduces the node feature energy from equation 11, which causes over-smoothing as shown by Wu et al. (2019); Wang et al. (2019) (as discussed in the main paper), here, we also show that the same behaviour is evident with Citeseer and Pubmed datasets in Fig. 5.

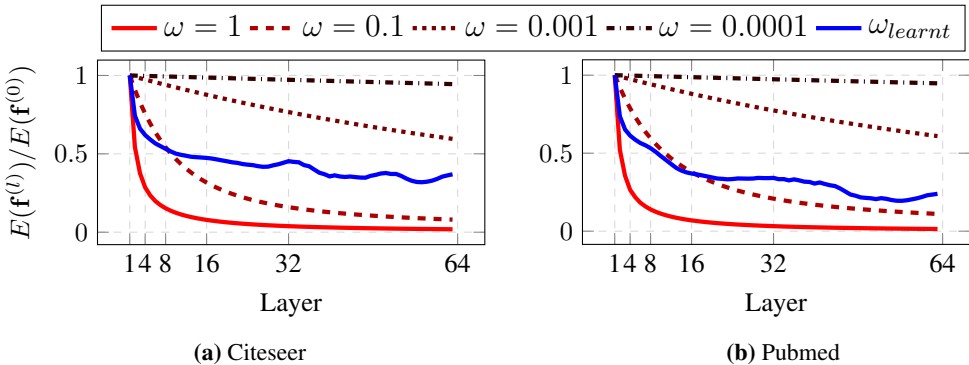

**(a)** Citeseer

**(b)** Pubmed

**Figure 5:** Node features energy at the $l$-th layer relative to the initial node embedding energy on Citeseer 5a and Pubmed 5b. $\omega$GAT controls the energy from Eq. equation 11 to avoid over-smoothing, while the baseline GAT with $\omega = 1$ reduce the energy to 0 and over-smooth.

## E  ARCHITECTURES IN DETAILS

We now elaborate on the specific architectures used in our experiments in Sec. 4. As noted in the main paper, all our network architectures consist of an opening (embedding) layer ($1 \times 1$ convolution), a sequence of $\omega$GNN (i.e., $\omega$GCN or $\omega$GAT) layers, and a closing (classifier) layer ($1 \times 1$ convolution). In total, we have two types of architectures – one that is based on GCN, for node classification tasks

reported in Tab. 10, and the other for the graph classification task which is based on Xu et al. (2019) and is reported in Tab. 11. Throughout the following, we denote by $c_{in}$ and $c_{out}$ the input and output channels, respectively, and $c$ denotes the number of features in hidden layers (which is a reported in Appendix F). We initialize the embedding and classifier layers with the Glorot (Glorot & Bengio, 2010) initialization, and $\mathbf{K}^{(l)}$ from equation 2 is initialized with an identity matrix of shape $c \times c$. The initialization of $\Omega^{(l)}$ also starts from a vectors of ones. We note that our initialization yields a standard smoothing process, which is then adapted to the data as the learning process progresses, and if needed also changes the process to a non-smoothing one by the means of mixed-signs, as discussed earlier and specifically in Theorem. 2. We denote the number of $\omega$GNN layers by $L$, and the dropout probability by $p$. The main difference between the two architectures are as follows. First, for the graph classification we use the standard add-pool operation as in GIN (Xu et al., 2019) to obtain a global graph feature. Second, we follow GIN and in addition to the graph layer (which is $\omega$GNN in our work), we add batch normalization (denoted by BN), $1 \times 1$ convolution and a ReLU activation past each graph layer.

**Table 10:** The architecture used for node classification and inductive learning.

| Input size | Layer | Output size |
|---|---|---|
| $n \times c_{in}$ | $1 \times 1$ Dropout(p) | $n \times c_{in}$ |
| $n \times c_{in}$ | $1 \times 1$ Convolution | $n \times c$ |
| $n \times c$ | ReLU | $n \times c$ |
| $n \times c$ | $L \times \omega$GNN layers | $n \times c$ |
| $n \times c$ | $1 \times 1$ Dropout(p) | $n \times c$ |
| $n \times c$ | $1 \times 1$ Convolution | $n \times c_{out}$ |

**Table 11:** The architecture used for graph classification.

| Input size | Layer | Output size |
|---|---|---|
| $n \times c_{in}$ | $1 \times 1$ Convolution | $n \times c$ |
| $n \times c$ | ReLU | $n \times c$ |
| $n \times c$ | $L \times [\, \omega$GNN , BN, $1 \times 1$ Convolution, ReLU $]$ | $n \times c$ |
| $n \times c$ | $1 \times 1$ Add-pool | $1 \times c$ |
| $1 \times c$ | $1 \times 1$ Convolution | $1 \times c$ |
| $1 \times c$ | $1 \times 1$ Dropout(p) | $1 \times c$ |
| $1 \times c$ | $1 \times 1$ Convolution | $1 \times c_{out}$ |

## F  HYPER-PARAMETERS DETAILS

We provide the selected hyper-parameters in our experiments. We denote the learning rate of our $\omega$GNN layers by $LR_{GNN}$, and the learning rate of the $1 \times 1$ opening and closing as well as any additional classifier layers by $LR_{oc}$. Also, the weight decay for the opening and closing layers is denoted by $WD_{oc}$. We denote the $\omega$ parameter learning rate and weight decay by $LR_{\omega}$ and $WD_{\omega}$, respectively. $c$ denotes the number of hidden channels. In the case of $\omega$GAT, the attention head vector $\mathbf{a}$ are learnt with the same learning rate as $LR_{GNN}$ and $WD_{GNN}$.

### F.1  SEMI-SUPERVISED NODE CLASSIFICATION

The hyper-parameters for this experiment are summarized in Tab. 12.

### F.2  FULL-SUPERVISED NODE CLASSIFICATION

The hyper-parameters for this experiment are summarized in Tab. 13. The number of layers used in Tab. 3 are mentioned in brackets in the table. For Ogbn-arxiv and Actor from Tab. 4, 8 layer $\omega$GCN and $\omega$GAT were employed.

**Table 12:** Semi-supervised node classification hyper-parameters.

| Architecture | Dataset | $LR_{GNN}$ | $LR_{oc}$ | $LR_{\omega}$ | $WD_{GNN}$ | $WD_{oc}$ | $WD_{\omega}$ | $c$ | $p$ |
|---|---|---|---|---|---|---|---|---|---|
| $\omega$GCN | Cora | 0.01 | 0.01 | 0.01 | 1e-4 | 8e-5 | 2e-4 | 64 | 0.6 |
| | Citeseer | 1e-4 | 0.005 | 0.005 | 1e-5 | 5e-6 | 2e-4 | 256 | 0.7 |
| | Pubmed | 0.001 | 5e-4 | 0.005 | 2e-4 | 1e-4 | 1e-4 | 256 | 0.5 |
| $\omega$GAT | Cora | 0.01 | 0.01 | 0.005 | 1e-5 | 1e-5 | 1e-5 | 64 | 0.6 |
| | Citeseer | 0.005 | 0.005 | 0.001 | 1e-4 | 1e-5 | 1e-4 | 256 | 0.7 |
| | Pubmed | 0.005 | 0.001 | 0.05 | 4e-5 | 1e-5 | 1e-4 | 256 | 0.5 |

### F.3 INDUCTIVE LEARNING

The hyper-parameters for the inductive learing on PPI are listed in Sec. 4.3 in the main paper, and are the same for $\omega$GCN and $\omega$GAT.

**Table 13:** Full-supervised node classification hyper-parameters.

| Architecture | Dataset | $LR_{GNN}$ | $LR_{oc}$ | $LR_{\omega}$ | $WD_{GNN}$ | $WD_{oc}$ | $WD_{\omega}$ | $c$ | $p$ |
|---|---|---|---|---|---|---|---|---|---|
| $\omega$GCN | Cora | 0.01 | 0.05 | 0.005 | 0.01 | 1e-4 | 1e-4 | 64 | 0.5 |
| | Citeseer | 0.001 | 0.08 | 0.005 | 0.005 | 1e-4 | 0 | 64 | 0.5 |
| | Pubmed | 0.005 | 0.005 | 0.01 | 0.003 | 5e-5 | 0.01 | 64 | 0.5 |
| | Chameleon | 1e-4 | 0.005 | 5e-4 | 1e-4 | 1e-4 | 1e-5 | 64 | 0.5 |
| | Actor (Film) | 0.05 | 0.01 | 0.05 | 1e-4 | 1e-4 | 1e-5 | 64 | 0.5 |
| | Cornell | 0.01 | 0.05 | 0.01 | 0.005 | 1e-4 | 0 | 64 | 0.5 |
| | Texas | 0.08 | 0.08 | 0.005 | 0.005 | 5e-4 | 0 | 64 | 0.5 |
| | Wisconsin | 0.001 | 0.05 | 0.005 | 1e-4 | 3e-4 | 3e-4 | 64 | 0.5 |
| | Ogbn-arxiv | 0.01 | 0.01 | 0.01 | 0 | 0 | 0 | 256 | 0 |
| $\omega$GAT | Cora | 0.001 | 0.01 | 0.05 | 0.001 | 5e-4 | 0 | 64 | 0.5 |
| | Citeseer | 0.005 | 0.05 | 0.03 | 0.005 | 5e-4 | 0.001 | 64 | 0.5 |
| | Pubmed | 0.05 | 0.005 | 0.005 | 0.003 | 1e-6 | 0.003 | 64 | 0.5 |
| | Chameleon | 0.005 | 0.005 | 3e-4 | 5e-4 | 5e-4 | 1e-5 | | |
| | Actor (Film) | 0.05 | 0.01 | 0.01 | 5e-4 | 0.001 | 4e-4 | 64 | 0.5 |
| | Cornell | 0.001 | 0.01 | 0.005 | 1e-4 | 1e-5 | 0 | 64 | 0.5 |
| | Texas | 1e-4 | 0.02 | 0.05 | 5e-4 | 5e-4 | 0 | 64 | 0.5 |
| | Wisconsin | 0.01 | 0.05 | 0.005 | 0.001 | 5e-4 | 0 | 64 | 0.5 |
| | Ogbn-arxiv | 0.01 | 0.01 | 0.01 | 0 | 0 | 0 | 256 | 0 |

### F.4 GRAPH CLASSIFICATION

The hyper-parameters for the graph classification experiment on TUDatasets are reported in Tab. 14. We followed the same grid-search procedure as in GIN (Xu et al., 2019). In all experiment, a 5 layer (including the initial embedding layer) $\omega$GCN and $\omega$GAT are used, similarly to GIN.

### F.5 ABLATION STUDY

In this experiment we used the same hyper-parameters as reported in Tab. 12.

## G RUNTIMES

Following the computational cost discussion from Sec. 2.4 in the main paper, we also present in Tab. 15 the measured training and inference times of our baselines GCN and GAT with 2 layers, where we see that indeed the addition of $\omega$ per layer and channel requires a negligible addition of time, at the return of a significantly more accurate GNN. We note that further accuracy gain can be achieved

**Table 14:** Graph classification hyper-parameters. BS denoted batch size.

| Architecture | Dataset | $LR_{GNN}$ | $LR_{oc}$ | $LR_\omega$ | $WD_{GNN}$ | $WD_{oc}$ | $WD_\omega$ | $c$ | $p$ | BS |
|---|---|---|---|---|---|---|---|---|---|---|
| $\omega$GCN | MUTAG | 0.01 | 0.01 | 0.01 | 0 | 0 | 0 | 32 | 0 | 32 |
| | PTC | 0.01 | 0.01 | 0.01 | 0 | 0 | 0 | 32 | 0 | 32 |
| | PROTEINS | 0.01 | 0.01 | 0.01 | 0 | 0 | 0 | 32 | 0 | 128 |
| | NCI1 | 0.01 | 0.01 | 0.01 | 0 | 0 | 0 | 32 | 0.5 | 32 |
| | NCI109 | 0.01 | 0.01 | 0.01 | 0 | 0 | 0 | 32 | 0 | 32 |
| $\omega$GAT | MUTAG | 0.01 | 0.01 | 0.01 | 0 | 0 | 0 | 32 | 0 | 32 |
| | PTC | 0.01 | 0.01 | 0.01 | 0 | 0 | 0 | 32 | 0 | 128 |
| | PROTEINS | 0.01 | 0.01 | 0.01 | 0 | 0 | 0 | 32 | 0 | 128 |
| | NCI1 | 0.01 | 0.01 | 0.01 | 0 | 0 | 0 | 32 | 0.5 | 128 |
| | NCI109 | 0.01 | 0.01 | 0.01 | 0 | 0 | 0 | 32 | 0.5 | 32 |

when adding more $\omega$GNN layers as reported in Tab. 2 in the main paper. However, since GCN and GAT over-smooth, the comparison here is done with 2 layers, where the highest accuracy is obtained for the baseline models.

**Table 15:** Training and inference GPU runtimes [ms] on Cora.

| Runtime | GCN | GAT | $\omega$GCN (Ours) | $\omega$GAT (Ours) |
|---|---|---|---|---|
| Training | 7.71 | 14.59 | 7.79 | 14.95 |
| Inference | 1.75 | 2.98 | 1.88 | 3.09 |
| Accuracy (%) | 81.1 | 83.1 | 82.6 | 83.4 |

# H ABLATION STUDY USING $\omega$GAT

To complement our ablation study on $\omega$GCN in Sec. 4.5 in the main paper, we perform as similar study on $\omega$GAT. Here, we show in Tab. 16, that indeed the single $\omega$ variant, dubbed $\omega$GAT$_G$ does not over-smooth, and that by allowing the greater flexibility of a per-layer and per layer and channel of our $\omega$GAT$_{PL}$ and $\omega$GAT, respectively, better performance is obtained.

**Table 16:** Ablation study on $\omega$GAT.

| Data. | Variant | Layers | | | | | |
| | | 2 | 4 | 8 | 16 | 32 | 64 |
|---|---|---|---|---|---|---|---|
| Cora | $\omega$GAT$_G$ | 83.3 | 83.3 | 83.4 | 83.6 | 83.7 | 83.9 |
| | $\omega$GAT$_{PL}$ | 83.4 | 83.5 | 83.8 | 84.0 | 84.1 | 84.0 |
| | $\omega$GAT | 83.4 | 83.7 | 84.0 | 84.3 | 84.4 | 84.8 |
| Cite. | $\omega$GAT$_G$ | 71.5 | 71.8 | 71.9 | 72.2 | 72.4 | 72.9 |
| | $\omega$GAT$_{PL}$ | 72.1 | 72.3 | 72.4 | 72.8 | 73.1 | 73.2 |
| | $\omega$GAT | 72.5 | 73.1 | 73.3 | 73.5 | 73.9 | 74.0 |
| Pub. | $\omega$GAT$_G$ | 80.0 | 80.2 | 80.3 | 80.5 | 80.6 | 80.9 |
| | $\omega$GAT$_{PL}$ | 80.0 | 80.4 | 80.7 | 81.1 | 81.2 | 81.4 |
| | $\omega$GAT | 80.3 | 81.0 | 81.2 | 81.3 | 81.5 | 81.8 |

**Table 17:** Semi-supervised node classification test accuracy 100 random train-val-test splits.

| Method | Cora | Citeseer | Pubmed |
|---|---|---|---|
| GCN (Kipf & Welling, 2016) | 81.5 | 71.9 | 77.8 |
| GAT (Veličković et al., 2018) | 81.8 | 71.4 | 78.7 |
| MoNet (Monti et al., 2017) | 81.3 | 71.2 | 78.6 |
| GRAND-l (Chamberlain et al., 2021) | 83.6 | 73.4 | 78.8 |
| GRAND-nl (Chamberlain et al., 2021) | 82.3 | 70.9 | 77.5 |
| GRAND-nl-rw (Chamberlain et al., 2021) | 83.3 | 74.1 | 78.1 |
| GraphCON-GCN (Rusch et al., 2022) | 81.9 | 72.9 | 78.8 |
| GraphCON-GAT (Rusch et al., 2022) | 83.2 | 73.2 | 79.5 |
| GraphCON-Tran (Rusch et al., 2022) | 84.2 | **74.2** | 79.4 |
| $\omega$GCN (ours) | **84.5** | 73.8 | **82.9** |
| $\omega$GAT (ours) | 84.3 | 73.6 | 82.6 |

## I  STATISTICAL SIGNIFICANCE OF SEMI-SUPERVISED NODE CLASSIFICATION RESULTS

Throughout our semi-supervised node classification experiment in Sec. 4 on Cora, Citeseer and Pubmed, the standard split from Kipf & Welling (2016) was considered, to a direct comparison with as many as possible methods. However, since this result reflects the accuracy from a single split, we also repeat this experiment with 100 random splits as in Chamberlain et al. (2021) and compare with applicable methods that also conducted such statistical significance test. In Tab. 17, we report our obtained accuracy on Cora, Citeseer and Pubmed. It is possible to see that in this experiment our $\omega$GCN and $\omega$GAT outperform or obtain similar results compared with the considered methods, which further highlight the performance advantage of our method.

## J  THE LEARNT $\vec{\omega}$

One of the main advantages of our method in Sec. 2 is that our method is capable of learning both smoothing and sharpening propagation operators, which cannot be obtained in most current GNNs. In Fig. 6 we present the actual $\{\omega^{\vec{(l)}}\}_{l=1}^{L}$ as a matrix of size $L \times c$ that was learnt for two dataset of different types—with high and low homophily score (as described in Pei et al. (2020)). Namely, the Cora dataset with a high homophily score of 0.81, and the Texas dataset with a low homophily score of 0.11 (i.e., a heterophilic dataset). We see that a homophilic dataset like Cora, the network learnt to perform diffusion, albeit in a controlled manner, and not to simply employ the standard averaging operator $\tilde{\mathbf{P}}$. We can further see that for a heterophilic dataset the ability to learn contrastive (i.e., sharpening) propagation operators in addition to diffusive kernels is beneficial, and is also reflected in our results in Tab. 3, where a larger improvement is achieved in datasets like Cornell, Texas and Wisconsin, which have low homophily scores (Rusch et al., 2022).

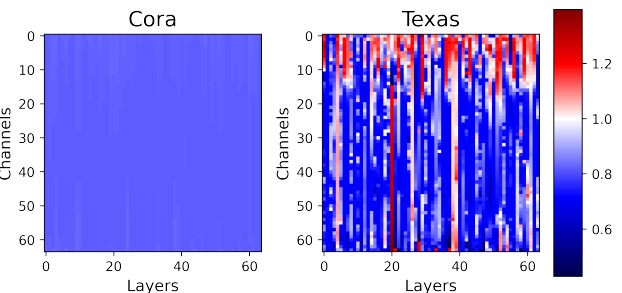

**Figure 6:** The learnt $\vec{\omega} \in \mathbb{R}^{64 \times 64}$ of $\omega$GCN with 64 layers (x-axis) and 64 channels (y-axis) for Cora (homophilic) and Texas (heterophilic) datasets. Smoothing operators appear in blue, while sharpening operators appear in red. White entries are obtained for $\omega = 1$.

