# OpenReview forum: "$\omega$GNNs: Deep Graph Neural Networks Enhanced by Multiple Propagation Operators"
_ICLR.cc/2023/Conference — Submitted to ICLR 2023_

### Official Review · Reviewer_shzp · 2022-10-22

**Confidence:** 4
**Correctness:** 2
**Technical Novelty And Significance:** 2
**Empirical Novelty And Significance:** 3
**Recommendation:** 5

**Clarity, Quality, Novelty And Reproducibility:**

The paper is clearly written with the problem well-defined and techniques precisely described.

My main concern on the quality of the paper is in its theoretical proof. See above.

I think the design is not entirely novel. As mentioned above, both layer-wise and channel-wise scaling of the propagation matrices have been proposed in the literature.

The experiments include reasonable amount of details on the setting and hyperparametes. The authors also provide source code in the supplementary materials.

**Strength And Weaknesses:**

## Summary of strengths

+ The paper is well-structured and easy to read. The motivation is clearly stated and the problem is well defined.
+ The overall architecture design is reasonable.
+ Extensive experiments have been performed on many datasets and across different tasks.

## Summary of weaknesses

- Theoretical proof is problematic. The derivation is based on an improper discretization process that results in the residue error to accumulate.
- The significance of the theoretical results is limited. First of all, it only discusses the basic version of omega-GNN with shared omega across channels. Further understanding is needed regarding the channel-wise scaling. Secondly, the analysis only discusses the prevention of oversmoothing. Many existing works can already address oversmoothing and it is not clear what are the advantages brought by omega-GNN.
- Scaling the propagation matrix either channel-wise or layer-wise has already been proposed in the literature.
- For experiments, most baselines are relatively old. Some more recent SOTA methods are missing. e.g., [a] which also addresses oversmoothing.

## Detailed comments

### Proof of Theorem 1 and Corollary 1

First, there is no guarantee that the Dirichlet energy of a multi-layer omega-GNN can converge to a target energy level, $E_{opt}$. It is true that Equation 16 defines a gradient descent operation with the Dirichlet energy as the minimization objective. However, iteration by Equation 16 is based on a constant step size omega, which is independent of both $E_{opt}$ and the iteration number (i.e., layer number). It is likely that $E(l+1) < E_{opt} < E(l)$ and thus there does not exist a layer $l$ to exactly achieve $E_{opt}$.

Second, the process of converting discrete form of Eq. 16 to the continuous form of Eq. 17, and then converting back to the discrete form of Eq. 20 is problematic. The discretization process introduces errors depending on omega, and such error accumulates with the layer number. Eventually, the second order term accumulated across the full GNN becomes a first order term and is not negligible.

Specifically, let $M = \tilde{D}^{-1/2}L\tilde{D}^{-1/2}$. Then Eq. 20 becomes

$f(t_{l}) = exp(-\omega M)f(t_{l-1}) = exp(-L\omega M)f(t_0)$

The target $\omega$ value is derived by setting $L\omega=T$.

However, if we look at the Taylor expansion:

$f(t_{l+1}) = (I-\omega M)f(t_l) + O(\omega^2) =  (I-\omega M)^2 f(t_{l-1}) + (I-\omega M)O(\omega^2) + O(\omega^2)= ...$

When the recursion is applied from $f(t_L)$ until $f(t_0)$, there will be $L$ terms of $O(\omega^2)$. Since $L=T/\omega$, then $L\cdot O(\omega^2) = O(\omega)$ and the residue error becomes non-negligible due to the recursive expansion.

Therefore,  Theorem 1 and Corollary 1 are problematic.

### Related works with similar ideas

* The idea of layerwise scaling the propagation matrix is known. Could you please discuss how omega-GNN improves methods such as GCNII?
* Channel-wise scaling the propagation matrix is also known. For example, in [b]. Please discuss the relation between omega-GNN and [b].
* Oversmoothing is a classic issue and there have been many solutions proposed, from many different perspectives. In addition to the ones in Sec 3, [a] also provably avoids oversmoothing without changing the GCN architecture. The theoretical analysis presented in the paper mostly focuses on the oversmoothing perspective. Could you please illustrate the advantages of addressing oversmoothing via omega-GNN compared with the many existing solutions?

### Experiments

Thanks for evaluating omega-GNN with many graphs and baselines. It would be more convincing if more recent SOTA methods (e.g., [a]) are included.

Figure 4 does not really demonstrate the similarity between solid and dashed lines. I'm not sure if it can really validate the theorems.


## References

[a] Decoupling the depth and scope of graph neural networks. In NeurIPS 2021.

[b] How Powerful are Spectral Graph Neural Networks. In ICML 2022.

**Summary Of The Paper:**

This paper presents a new GNN architecture, omega-GNN, which learns a filtering coefficient for each feature channel and each layer. The introduced omega parameter scales the graph convolution and thus enables "sharpening" operation in addition to the "smoothing" performed by traditional GNNs. By properly setting the omega parameters, the authors claim that omega-GCN does not oversmooth. A more expressive variant performing channel-wise scaling has been proposed and instantiated on the GCN and GAT backbones. Extensive experiments have been performed on both node-level and graph-level tasks to show the accuracy improvements.

**Summary Of The Review:**

In summary, I think this is a well-written paper with reasonable architecture design. The idea bears similarity with a few existing works. The theoretical analysis seems problematic. And it would be better to also (informally) analyze the benefits beyond addressing oversmoothing -- a phenomenon already with many solutions.

Overall, in its current form, I think the paper is still below the bar of acceptance.

---

> ### Author Response · Authors · 2022-11-17
> **Response (part 1)**
>
> We thank the reviewer for all the detailed comments and interest in our proofs. Below, we address your concerns.
>
> **Regarding proofs of Theorem 1 and Corollary 1.1:** We thank the reviewer for the thorough reading of our proof. The reviewer raises a correct point about the first order derivation in our proof, but we humbly think that the reviewer's conclusion is not accurate. The analysis of the reviewer shows that the accumulation of the per-layer error results in an $O(\omega)$ term. This analysis actually does show that we have convergence of the discrete steps (layers) and the continuous integration. In Theorem 1, we show that the parameter $\omega$ will read $\omega = \frac{T}{L}$ where $T$ is fixed. Therefore, at the limit where **$L\rightarrow \infty$, it holds that $\omega \rightarrow 0$ and the discrete steps converge to the continuous function in time**. That is in fact a classical result in numerical integration methods, and we will add it to the revised manuscript. Also note that the matrix $(I-\omega\mathbf{M})$ also plays a role in reducing the error as its spectral radius is bounded by 1. That is, for $\omega < \frac{2}{\rho(\textbf{M})}$ we have that $\rho(I-\omega\mathbf{M}) < 1$. Hence, taking the powers of this matrix is guaranteed to reduce the accumulated error. That is also a classical result of numerical time integration, that leads to the CFL stability condition (Numerical Solution of PDEs, Morton and Mayers). In fact, our situation is equivalent to the Forward Euler method for integrating the heat equation, which, is known to converge given that the time steps are small enough. Despite all our assumptions, Fig. 4 validates Theorem 1 AND Corollary 1.1 regarding the per-layer case using real experiments, and demonstrates the $O(\omega)$ error.
>
> Furthermore, following the reviewers note **"Equation 16 is based on a constant step size omega"** we would like to stress that we do not assume an arbitrary constant step size, but rather one that is **based on the learnt $\omega$**. Therefore, it is dependent on the number of layers, because we discussed above and as shown in Fig. 4 in our ablation study, the learnt values of $\omega$ vary according to the number of layers in the network.
>
> Also, to address your question of whether the optimal energy can be "between" the energies of subsequent layers, we added a simple experiment that starts from node features with an initial Dirichlet energy, and we show that our $\omega$GNNs can achieve some different node features with a target energy (whether smaller or higher than the initial energy) using the $\omega$ parameter only. This shows that indeed if the energy is known our $\omega$GNNs can obtain it. When working with a real dataset induced by a real problem, as discussed above, it is safe to assume that some energy exists (which we may not know), but it is guided by the considered objective function (such as the Cross-Entropy loss).

---

> > ### Author Response · Authors · 2022-11-17
> > **Response (part 2)**
> >
> > **Regarding an analysis of the per-channel $\omega$:** We agree that conducting a theoretical study of the channel-wise scaling is important. Having the channel-wise scaling significantly complicates the proof, because in this case we also need to include the channel mixing operator (1x1 convolution). The influence of this operator was only recently analysed in the paper [c] below, from a spectral perspective, and combining the mixing operator into our theory is not trivial and is a subject of future research. We believe that our global and per-layer $\omega$ theory is valuable as is for researchers in the GNN community to design architectures that are inherently resilient to over-smoothing.
> >
> > $[$c$]$ Di Giovanni et al. Graph Neural Networks as Gradient Flows: understanding graph convolutions via energy.
> >
> > **Regarding other over-smoothing methods:** In the related work section we cite various methods that address over-smoothing either through regularization, keeping the original features, or by employing rather complicated architectures. While the approaches are beneficial and work well for the current data sets, they come as corrections to the over-smoothing problem, and not as an inherent solution with a *proper analysis of how this solution behaves*. Furthermore, we also elegantly employ multiple propagation operators into the architecture which is the real advantage of our work in terms of yielding the highest accuracy. The simplicity of $\omega$GNN also allows to achieve multiple propagation operators through point-wise scaling only. Those propagation operators can yield either an identity mapping, a smoothing operator, or a sharpening operator. which will then be further combined and mixed using MLPs.
> >
> > Specifically, to address the question about the difference between our $\omega$GNNs and GCNII, we highlight three key differences:
> > 1. Our $\omega$GNNs learn the appropriate type (identity, smooth, or sharpen) and amount of propagation, while GCNII performs a diminishing smoothing process controlled by pre-defined hyper-parameters.
> >
> > 2. Our $\omega$GNNs show a significant performance (accuracy) compared to GCNII.
> >
> > 3. While in GCNII there is in a single pre-defined smoothing propagation operator at each layer, our $\omega$GNNs learn a propagation operator per channel and layer.
> >
> > **Regarding other works:** We were not aware of the recennt work [b] before the submission and we will cite and compare with in the revised version.
> > In the work [b], we notice a discussion in Section 6.3 regarding difficulties in the optimization, for which the authors use a Polynomial Coefficient Decomposition to solve it. In our $\omega$GNNs we did not see any optimization issues, and our method yields similar or better accuracy, as shown below.
> >
> > | Method         | Cora  | Citeseer | Pubmed | Chameleon | Actor | Squirrel | Cornell | Texas |
> > |----------------|-------|----------|--------|-----------|-------|----------|---------|-------|
> > | JacobiConv [b] | 88.98 | **80.78**    | 89.62  | **74.20**     | **41.17** | 57.38    | **92.95**   | 93.44 |
> > | wGCN (ours)    | **89.30** | 77.88    | 90.45  | 70.02     | 38.94 |**59.41**    | 91.35   | 94.05 |
> > | wGAT (ours)    | 89.25 | 78.01    | **90.65**  | 72.23     | 38.64 | 58.96    | 91.62   | **94.59** |
> >
> >
> > **Regarding comparison with [a]:** In our comparison we compare both with older popular methods like GCN and GAT, and we also compare to newer works like GCNII (ICML 2020), GRAND (ICML 2021), EGNN (NeurIPS 2021), PDE-GCN (NeurIPS 2021), CIN (NeurIPS 2021), SIN (ICML 2021), SD (NeurIPS 2022), GSN (TPAMI 2022). We appreciate the additional reference by the reviewer which we will compare to in our revised paper.
> >
> > Specifically, on ogbn-arxiv [a] obtains an accuracy of 72.74 \% while our $\omega$GCN obtains 73.02 \%. We will add [a] to the comparisons in the revised version.
> >
> > **Regarding the qualitative advantage and difference between [a] and our $\omega$GNNs:** While both of the methods show a method to avoid over-smoothing using different mechanisms, we would like to note that as discussed earlier our $\omega$GNNs provide more than that. Specifically, we allow to learn multiple propagation operators of different types (identity, smoothing, sharpening) and mix them. This is a crucial point that can be seen from our experiments.
> >
> >
> > **Regarding summary and novelty:** In our rebuttal above we addressed your concerns and we hope that you find them satisfactory. We hope to hear back from you.

---

> > > ### Comment · Reviewer_shzp · 2022-11-21
> > > **Reviewer's response**
> > >
> > > Thank you for providing such a detailed rebuttal and I appreciate the additional experimental comparisons.
> > >
> > > Please integrate the above rebuttal into your next revision.
> > >
> > > I agree that GCNII lacks the mechanism of sharpening. Thus, $\omega$GNN is a valuable extension to GCNII.
> > >
> > > For JacobiConv [b], it seems that $\omega$GNN does not achieve significant performance gain. The main advantage of $\omega$GNN, as mentioned in the rebuttal, may be the easier optimization procedure. However, both $\omega$GNN and JacobiConv are designed for small graphs only (as they operate on the full graph Laplacian), and so it is unclear how significant it is to have an easier optimization procedure.
> > >
> > > For SHADOW-GNN [a], it seems that $\omega$GNN marginally improve the accuracy on ogbn-arxiv. However, it is also important to notice that SHADOW-GNN is designed for large scale graphs and may have the advantages in terms of scalability and efficiency.
> > >
> > > I believe each of the mentioned methods has its own pros and cons in terms of performance. I would suggest the authors carefully discuss them in the next revision of the paper. This will allow the readers to reasonably judge the significance of the experimental performance of the proposed method.

---

> > > > ### Author Response · Authors · 2022-11-23
> > > > **Discussion**
> > > >
> > > > We thank the reviewer for the interesting discussion and are happy to learn that the reviewer has accepted some of our responses. Below, we address the remaining issues.
> > > >
> > > >
> > > > 1. We agree that JacobiConv obtains impressive accuracy, as seen in the comparison in our response. It seems to us, from this comparison, that there is not a clear better method, at least experimentally, and both are good. On the qualitative side, our approach is based on a minimal change to GNNs that conform to Equation (1), targeting spatial GNNs, while JacobiConv is spectral-based.
> > > >
> > > > 2. In your review, we were asked to compare to this method, which we did, and showed that our $\omega$GCN outperforms SHADOW-GNN (even if not by a large margin). We appreciate the comment about the computational efficiency of SHADOW-GNN, which we will add to the discussion of the revised paper. But, we would like to stress that computational efficiency is not the focus of our $\omega$GNN (although they do not cost more computations than the baseline GCN/GAT as shown in Table 15 in our paper).
> > > >
> > > > If the reviewer still feels that our paper does not merit a change of score, we would be happy to know why and further discuss it.

---

> > ### Comment · Reviewer_shzp · 2022-11-21
> > **Reviewer's response**
> >
> > Thank you for providing a detailed rebuttal! I have read the response carefully, and here are my thoughts after the authors' clarification.
> >
> > First, regarding the first order error: I agree that in the limit of infinite layers, the analysis of discrete layers reduces to the continuous integration. However, my original comment still holds, that the first order error term is missing in the proof and it only vanishes for infinite number of layers.
> >
> > The same reasoning applies to the attainability of the optimal energy. The proposed model can only attain an arbitrary positive optimal energy with an arbitrarily small step size. In other words, the strict requirement is that the model has infinite number of layers.
> >
> > Summarizing the above two points, the theoretical results only apply in the case of an infinitely deep GNN. There is no guarantee for a realistic GNN. This makes the theorem less significant.
> >
> > In addition, I agree with cjVH that the oversmoothing is "hidden" rather than resolved in the proposed infinite-layer setup. I think the design is to increase the number of layers by proportionally decreasing the step size so that it simply achieves the same smoothing effect of a shallower GNN with larger step size. In this case, I wonder what the real benefit is to increase the depth (especially considering that the analysis assumes a linear model without activation).

---

> > > ### Author Response · Authors · 2022-11-23
> > > **Discussion**
> > >
> > > We thank the reviewer for responding to our rebuttal. Below we address the remaining issues:
> > >
> > >
> > > 1. The error term shows the difference between the continuous and discrete integration. Any error term (also second order and beyond) will not entirely vanish unless we have a finite number of layers. However, having that error does not take away from the analysis that explains the way the mechanism works. The input sample (available nodes) is arbitrary, and that causes much more variation in the energy than this error term for a reasonable number of layers. We note that we do not choose $\omega$. We optimize for it, and our analysis agrees (up to first order) with the optimization result that also introduces an error (a local minimum).
> > >
> > >
> > > 2. Regarding optimal energy: Indeed, the analysis uses the analytical result of continuous integration and holds up to a first-order error. But, since the step size $\omega$ is learnt and continuous, for a deterministic set of examples one can reach any desired energy through the choice of $\omega$. This is a property of the continuity in $\omega$. Regardless, this is a learning problem that generalizes over many examples. We do not think there is a natural way to define an exact energy value to the examples in the datasets. Further, such a theoretical optimal energy value will also vary on the embedding of the input. Instead, in our Theorem, we wish to analyze, and in Figure 4 we demonstrate, the overall behaviour of our method. Again -  we optimize for $\omega$, and our analysis agrees (up to first order) with the optimization result (the optimization is another source of error in the form of a local minimum).
> > >
> > >  To summarize, Figure 4 demonstrates our theorem for a reasonable number of layers. There is no need for an infinite number of layers, and reaching exact energy is quite meaningless in a stochastic learning problem.
> > >
> > > 3. Re hiding over-smoothing: We would like to kindly stress that we do not hide the over-smoothing problem in our $\omega$GNNs, because, if desired, $\omega$ can be learnt to be zero, rendering a simple 1x1 convolution that does not propagate information. Yet, we claim that smoothing is beneficial for obtaining good accuracy if it is **controlled** and **learnt** (i.e., learing the right amount of the smoothing), so that the network does not **over**-smooth. In most of our experiments, the best accuracy is obtained after 64 layers, and our energy plots show that the energy does not decay to zero. What we get in our analysis and its demonstration is the result of an optimization process. This is, in some sense, what the data "wants", and our $\omega$GNNs enable that because if the energy needs to be smaller, then we will learn to gravitate towards a smoothing network, and in case the energy needs to be higher, our $\omega$ parameter can gravitate the network towards a sharpening one. The benefit of an increased depth is in the more $1\times1$ convolutions and non-linearities in between the propagation layers, leading to a more complex network. Indeed, in many cases, the best accuracy is obtained with more layers in our experiments.
> > >
> > > If the reviewer still feels that our paper does not merit a change of score, we would be happy to know why and further discuss it.

---

### Official Review · Reviewer_cjVH · 2022-10-24

**Confidence:** 4
**Correctness:** 3
**Technical Novelty And Significance:** 4
**Empirical Novelty And Significance:** 4
**Recommendation:** 5

**Clarity, Quality, Novelty And Reproducibility:**

The paper is well-prepared, clearly written and easier to follow. Introducing a parameter into the graph convolution operator is novel and meaningful.

**Strength And Weaknesses:**

The strengths of the paper:

+ It is quite interesting to develop a parameterized graph convolution operator that is able to prevent the over-smooth issue and is able to implement variant operators.

+ The empirical evaluation is extensive and the experimental results show notable improvements.

The weaknesses of the paper:

- Note that the graph convolution operation in GCN is a step of gradient-based node updating to optimize an objective in label propagation. Here, a parameter $\omega$ introduced in Eq. (2), serves as a step size parameter. If the parameter $\omega$ is learnable, does it still converge in the perspective of understanding the graph convolution as optimizing a label propagation objective (e.g. Ma et al.'21)? Where does it converge to?

Ref.
[a] Yao Ma et al. 2021. A Unified View on Graph Neural Networks as Graph Signal Denoising.

- Note that, the parameter $\omega$ is a step size to control the degree of incorporating the graph convolution to modify the node feature. Of course, using a smaller $\omega$, the smoothing issue could be slower as shown in Fig.3. However, it does not resolve the smoothing issue at all. When using a smaller $\omega$, more layers of graph convolusions are needed to yield an equivalent node feature. Thus, it is NOT to avoid over-smoothing issue at all ---the over-smoothing issue is still hidding in the problem formulation!
Looking into the curve in Fig.3(b) for $\omega$GAT, it seems not showing a convergence. If many iteratons (i.e., layers) are conducted, does it converge or not?

- It is interesting to have a theorem to justify the convergence for the proposed methods. However, the Theorem 1 strongly relies on the assumption that there is some optimal Direchlet energy of the final feature map that satisfies $0 < E_{opt} (f^{(L)}) < E(f^{(0)})$. Does it really exist an optimal value that is strictly greater than 0? In such case, what is the condition on the property of the objective function or on the graph Laplacian? Is the implicit optimization problem a well-defined, in the sense that not having a trivial solution?



**Summary Of The Paper:**

The paper attempts to address the shortcomings in GNNs by modifying the propagation operator and thus propose an approach called $\omega$GNN, as two instances, $\omega$GCN and $\omega$GAT. It is justified that $\omega$GNN can prevent over-smoothing issue, and the parameterized propagation operator in $\omega$GNN enable variant operators to use. Extensive experiments are conducted and shown superior accuracy.


**Summary Of The Review:**

The paper is well-motivated and clearly written. The idea is interesting and novel. The empirical evaluation is extensive. Nevertheless, it is suspecious that the contribution is somewhat over-claimed. To be specific, the over-smoothing issue is just alleviated, but not resolved yet. For details, please refer the "weaknesses".

---

> ### Author Response · Authors · 2022-11-17
> **Response (part 1)**
>
> We thank the reviewer for the questions and comments. Below we address them one by one.
>
>
> **Regarding gradient step, convergence, and paper [a]**: We would like to thank you for the reference of [a] which we will cite and discuss in our revised paper. To be specific, [a] considers a graph denoising perspective coupled with the propagation rules of various GNNs like APPNP, GCN and GAT. The theoretical step size provided in [a] is coupled to an additional smoothness regularization term that does not exist in our work (our $\omega$GNNs do not require / assume further objectives apart from the basic task-driven loss like the cross-entropy loss). Therefore we cannot directly compare the value of $\omega$ to the theoretical value present in [a]. To address the question of the convergence of $\omega$ we split the answer into two parts:
> 1. In our ablation study (please see Fig. 4) we show the sum of $\omega$ for a varying number of layers. It can be seen that when learning a single $\omega$ (also dubbed as $\omega$GNN$_{G}$ in our paper), which is the counterpart of the model presented in [a], our Theorem 1 is empirically validated. That is, up to first order errors, the sum of $\omega$ remains almost constant. Please note that our Theorem 1 is also based on treating each GCN layer as a gradient step to decrease the Dirichlet Energy of the node features, similarly to [a].
> 2. After establishing that Theorem 1 holds and we understand how $\omega$ behaves as a function of the number of the parameters, we now approach the question whether $\omega$ converges. We now repeated the experiment presented in Fig. 4 with 100 random initializations. We found that the standard-deviation (std) on Cora for each #layers configuration (as we consider 2 to 64 layers) is as follows: $$[0.32, 0.27, 0.20, 0.13, 0.06, 0.008],$$ with mean sum $\omega$ value per layer of: $$[1.96, 2.11, 2.17, 2.38, 2.30, 2.28].$$ This result shows that our $\omega$ converges with similar settings to [a].
> Furthermore, in our original submission, in Appendix I, we provided the learnt $\vec{\omega}$ on a homophilic and a heterophilic datasets, to show the actual learnt values. The inspection of those values shows that while for homophilic datasets it may be sufficient to control the amount of smoothing, for heterophilic datasets it is important to also be able to learn and use mixed-sign operators like sharpening operators.
>
> **Regarding avoiding over-smoothing:** We thank your for the important question. We first would like to state that using our $\omega$GNN, the over-smoothing problem can be avoided altogether and we now explain why. As can be seen in Eq. (2), we separate between an identity mapping, and the actual feature propagation denoted by $(I-S)$ that is multiplied by $\omega$. Therefore, if $\omega=0$, please notice that *no propagation will occur*. Therefore, if desired, our $\omega$GNNs can learn to perform no smoothing at all. Additionally, it is important to note that **we do not claim that smoothing is not needed**. In fact, **we claim that smoothing is beneficial when controlled and properly learnt** (which is thoroughly discussed in our Introduction [Section 1] in Page 2, and in the Related Work section [Section 3] in Page 6, paragraph "Graph Neural Diffusion", and also in Appendix J.
> We hope that this point is clearer now and we are happy to further discuss it if you have more concerns/questions.
>
> **Regarding Fig. 3(b) and convergence:** We now expanded this experiment to 128 layers and we found that it converges after 64 layers. We updated Fig. 3 accordingly in our revised paper. Moreover, it is important to note that for GAT, we do not claim that the energy value necessarily converges (because its propagation operator can change at every layer and it is not symmetric by construction). Our claim is that the energy will not decrease to 0, thus preventing the over-smoothing problem.

---

> > ### Author Response · Authors · 2022-11-17
> > **Response (part 2)**
> >
> > **Regarding the value of the optimal energy:** We agree that this is an important aspect. First, we would like to establish that it is safe to claim that there exists some optimal energy and in particular that it is non-negative. The non-negativity follows immediately from the definition of the Dirichlet energy in Eq. (6). Second, let there be some optimal features $\textbf{f}^{opt}$. Those features have an associated optimal Dirichlet energy that is obtained by plugging $\textbf{f}^{opt}$ into Eq. (6).
> >
> > Hence, in general there exists some optimal energy value. Moreover, it is not reasonable to assume that the optimal energy equals to zero (i.e., $E(f^{opt})=0$). If the graph is connected, then having an optimal energy of zero means having the same feature to all the nodes, which cannot distinguish between different nodes for node classification (hence, how can it be optimal?). Also, it defeats the purpose of avoiding over-smoothing as discussed in the papers ["Measuring and relieving the over-
> > smoothing problem for graph neural networks from the topological view" (Chen et al., 2020)], ["A Note on Over-Smoothing for Graph Neural Networks" (Cai and Wang, 2020)]. We will add this discussion to our revised paper.
> >
> > Lastly, we address the concern of whether the optimal energy is necessarily lower than the initial features energy, i.e., $E(f^{opt}) \leq E(f^{(0)})$. We would like to state that indeed this is an assumption of Theorem 1 and we find it reasonable as it is a common assumption among many GNN models such as APPNP[Klicpera 2019], GDC [Gasteiger, 2019], GCNII [Chen, 2020], GRAND [Chamberlain, 2021],  PDE-GCN [Eliasof, 2021], GraphCON [Rusch, 2022]. This assumption is also made in in the reference [a] provided by the reviewer, which we will cite in our revised paper. Furthermore, in the case where the final, optimal energy is desired to be greater than the initial energy, i.e., $E(\textbf{f}^{opt}) \geq E(f^{(0)})$, our $\omega$GNNs have the freedom to learn sharpening operators through negative $\omega$ values to increase the energy. We will add this discussion to our revised paper.
> >
> >
> > **Regarding summary**: We thank you for your comments. In our response we addressed your concerns regarding overcoming over-smoothing. Specifically, we clarify the question whether our $\omega$GNNs only ease or completely avoid the over-smoothing problem. The answer is that our $\omega$GNNs do avoid this problem. We also discuss [a], and the address the question of convergence of $\omega$. We hope that you find our answers satisfactory and hope to hear from you.

---

> > > ### Comment · Reviewer_cjVH · 2022-11-20
> > > **Reply to the responses**
> > >
> > >
> > > Thanks for the authors carefull responses. The responses have addressed some concerns but not all of them.
> > >
> > > 1. Note that Theorem 1 relies on the assumption that there is some optimal Direchlet energy of the final feature map that satisfies $0 < E_{opt} (f^{(L)}) < E(f^{(0)})$. The reviewer does not think that there is always an optimal value that is strictly greater than 0 without taking into account of the proper of the graph Laplacian. To make the energy to zeros, there are at least two feasible ways: a) over-smoothed trivial solution; b) bad condition of the graph Laplacian, e.g, poor connecticity issue (i.e., algebraic connectivity is zero). What about the latter case is not clear.
> > >
> > > 2. The learned $\omega$, especially the negative components in $\omega$ are interesting. Nevertheless, when there are negative components in $\omega$, how can we interprete them from an optimization perspective? I guess some of the updatings are not "descent step", but somehow "ascent step"? What is the meaning or significance if we solve an optimization objective with mixed directions for updating? Is that a behaviour in optimization process to attempt into overfitting? While, it seems a right "direction" numerically to resolve the over-smoothing issue to increase the objective function, but more discussions to justify or interprete the rationale are needed.

---

### Official Review · Reviewer_UJFL · 2022-10-24

**Confidence:** 4
**Correctness:** 3
**Technical Novelty And Significance:** 2
**Empirical Novelty And Significance:** 2
**Recommendation:** 3

**Clarity, Quality, Novelty And Reproducibility:**

Clarity of writing is in my opinion quite good. The novelty is not as large as claimed, as noted in the strengths + weaknesses section. The authors should contrast their method with the other mixed-sign methods that they did not initially cite.

**Strength And Weaknesses:**

Strengths:
1. Simple approach that is easy to implement
2. Use of multiple omegas gives a flexible class of operators in an efficient manner
3. Experiments on both node and graph classification tasks, while other works often do one or the other.

Weaknesses:
1. Should cite [1], which analyzes oversmoothing using the Dirichlet energy.
2. Missing citations to other mixed sign propagations in GNNs, this is done by others besides Eliasof et al. 2022. See [2], [3], [7] for prominent examples. Moreover, this is also mentioned in [4]. Please compare and explain the differences.
3. Why would you not also run your method on the Squirrel dataset in Pei et al. 2020?
4. The datasets tested on for the most part have been noted to have a lot of issues. For instance, the Pei et al heterophily datasets were covered in [5] and Cora + Citeseer + Pubmed were covered in [6]. New datasets were introduced, which your method is clearly scalable enough to run. Further, many methods do quite well on PPI, and your gain is only 00.02 better than the next best method. Also, there are better baselines that outperform your method on ogbn-arxiv, such as Correct and Smooth [8] from 2020.


Other notes:
1. Should be $\vec \omega^{(l)} \in \mathbb{R}^c$ right before equation (3)

[1] Cai and Wang. A Note on Over-Smoothing for Graph Neural Networks. https://arxiv.org/abs/2006.13318

[2] Bo et al. Beyond Low-frequency Information in Graph Convolutional Networks. https://arxiv.org/abs/2101.00797

[3] Yan et al. Two Sides of the Same Coin: Heterophily and Oversmoothing in Graph Convolutional Neural Networks. https://arxiv.org/abs/2102.06462

[4] Di Giovanni et al. Graph Neural Networks as Gradient Flows: understanding graph convolutions via energy. https://arxiv.org/abs/2206.10991

[5] Lim et al. Large Scale Learning on Non-Homophilous Graphs: New Benchmarks and Strong Simple Methods. https://arxiv.org/abs/2110.14446

[6] Shchur et al. Pitfalls of Graph Neural Network Evaluation. https://arxiv.org/abs/1811.05868

[7] Yang et al. Diverse Message Passing for Attribute with Heterophily. NeurIPS 2021

[8] Huang et al. Combining Label Propagation and Simple Models Out-performs Graph Neural Networks. https://arxiv.org/abs/2010.13993






**Summary Of The Paper:**

Simple changes are proposed to overcome expressivity limits of GNN propagation. The changes involve new parameters $\omega$, which very naturally allow both smoothing-type propagations and sharpening-type propagations. Instantiations of the proposed changes in GCNs and GATs are empirically tested in both node and graph level tasks.




*Correctness
3

*Technical Novelty and Significance
2

*Empirical Novelty and Significance
2

*Flag for ethics review:
No

*Recommendation
3

*Confidence
4


**Summary Of The Review:**

It is good to see simple and reasonable changes to GNNs that can help with some of the known issues of these models. However, the exact novelty is unclear, as several other similar methods (some of which are quite prominent) have been proposed. Moreover, the choice of datasets is not great for showing significant empirical benefits.

---

> ### Author Response · Authors · 2022-11-17
> **Response (part 1)**
>
> We thank the review for the insightful comments and we now address them one by one.
>
> **Regarding citing [1]:** Thank you for the reference. We will cite and discuss [1].
>
> **Additional mixed-sign operators papers ([2], [3], [4], [7]):** We thank you for the additional references. We will incorporate those papers into our revised version and clearly distinguish the difference between them and our $\omega$GNNs.
>
> The biggest difference that is relevant to all of the aforementioned methods is that **our $\omega$GNNs offer per channel, per layer propagation operators** that also come with a minimal cost of the number of channels per layer and a simple scalar multiplication, while **most of the other methods propose a single propagation operator per layer**.
>
> Also, [2],[3],[7] propose an attention based propagation operator with a tanh/cosine activation function to obtain values between [-1,1]. These methods do not offer a direct parameterization of the learnt mixed-sign kernels as our $\omega$GNNs. Furthermore, these methods require a pair-wise product that is significantly more expensive and depends on the number of nodes and neighbourhood size.
>
>
> We now provide specific differences as follows (we number the differences according to the papers numbered by the reviewer):
>
> [2]. Offers an attention (pair-wise) computation to determine the propagation weight. This method requires more computations than ours and also comparing the results, our $\omega$GNNs achieves higher accuracy (please see the table below).
>
> [3]. This method computes the cosine distance for each edge of the graph to determine negative and positive interactions. Similar to [2], it requires the pair-wise computation which scales as the number of nodes and neighbourhood size of the graph. Our $\omega$GNNs do not require that for learning the $\omega$ weights and obtain better experimental results (please see the table below).
>
> [4]. This method offers an analysis and experimental effort from the perspective of the **mixing** weights perspective (i.e., 1x1 convolution), while our $\omega$GNNs offer an analysis from the perspective of the **propagation** (spatial) operator. Also, our experimental results provide higher accuracy (please see the table below).
>
> [7]. This method also offers an attention (pair-wise) computation to determine the propagation weight with slightly different parameterization (e.g., a different activation function) This method requires more computations than ours and also comparing the results, our $\omega$GNNs achieves higher accuracy (please see the table below).
>
> **Semi-supervised node classification**:
> | Method      | Cora | Citeseer | Pubmed |
> |-------------|------|----------|--------|
> | FAGCN [7]   | 84.1 | 72.7     | 79.4   |
> | wGCN (ours) | **85.9** | 73.3     | 81.1   |
> | wGAT (ours) | 84.8 | **74.0**     | **81.8**   |
>
> **Fully-supervised node classification**:
> | Method                       | Cora      | Citeseer  | Pubmed    | Chameleon | Actor (Film) | Squirrel  | Cornell   | Texas     | Wisconsin |
> |------------------------------|-----------|-----------|-----------|-----------|--------------|-----------|-----------|-----------|-----------|
> | GGCN [3]                     | 87.95     | 77.14     | 89.15     | 71.14     | 37.54        | 55.17     | 85.68     | 84.86     | 86.86     |
> | GRAFF [4]                    | 88.01     | 77.30     | 90.04     | 71.08     | 37.11        | 58.72     | 84.05     | 88.38     | 88.83     |
> | GRAFF NL [4]                 | 87.81     | 76.81     | 89.81     | 71.38     | 35.96        | 59.01     | 77.30     | 86.49     | 87.26     |
> | DMP (best of all models) [7] | 86.52     | 76.87     | 89.27     | 62.28     | 35.72        | 47.26     | 89.19     | 89.19     | 92.16     |
> | wGCN (ours)                  | **89.30** | 77.88     | 90.45     | 70.02     | **38.94**    | **59.41** | 91.35     | 94.05     | 92.35     |
> | wGAT (ours)                  | 89.25     | **78.01** | **90.65** | **72.23** | 38.64        | 58.96     | **91.62** | **94.59** | **92.94** |
>
>
> **Regarding experimenting with the Squirrel dataset:** Thank you for the suggestion. We now also experiment with squirrel (please see the table above).

---

> > ### Author Response · Authors · 2022-11-17
> > **Response (part 2)**
> >
> > **Regarding the chosen datasets:** We thank you for the useful references and constructive comment.
> >
> > In our experiments we wanted to be able to compare with as many recent methods as possible and therefore we used the popular datasets (which are also the datasets used in the references [2, 3, 4, 7] provided by the reviewer). We will also add a reference and discussion of [5] and [6] to our revised paper.
> >
> > We would like to add that to address the problems presented in [6], we included in our original submission an experiment with 100 random splits of Cora/Citeseer/Pubmed, in Appendix H. We also now compare with, cite, and discuss [8] in our revised paper. Our $\omega$GCN obtains 73.02\% accuracy, while [8] (Correct and Smooth) obtains 73.93\%. On other datasets like Cora/Citeseer/Pubmed, our $\omega$GNNs obtain 89.30/78.01/90.65 \% which is higher than [8] with 88.49/77.99/90.30 \%, respectively. It is also important to note that our $\omega$GNNs offer an extended contribution compared to [8] by allowing to learn both smoothing and non-smoothing propagation operators, and therefore also improving the obtained results on heterophilic datasets which are not considered in [8].
> >
> > **Regarding other notes / typos**: Thank you for the suggestion. We will incorporate it into our revised paper.
> >
> > **Regarding novelty and summary:** Dear reviewer, in our response we addressed each of your concerns regarding novelty and further highlighted the differences between our $\omega$GNNs and other methods like [2,3,4,7]. We also added an extensive quantitative comparison between our $\omega$GNNs and those methods. We also addressed your important comment regarding the choice of datasets and embraced your suggestion of also evaluating our method on the Squirrel dataset.
> > We hope that you find our response satisfactory and we would love to hear back from you.

---

> > > ### Comment · Reviewer_UJFL · 2022-12-06
> > > **response**
> > >
> > > It is good that the authors will incorporate these related works into their submission, but I still vote to reject. Their novelty and empirical contribution is not as great as they state in the paper. They do contrast against other mixed-sign methods in their response part 1, but these are not huge differences. For instance, per channel operators are a very simple generalization that have been done in other places.

---

### Official Review · Reviewer_ZYQh · 2022-11-01

**Confidence:** 3
**Correctness:** 3
**Technical Novelty And Significance:** 4
**Empirical Novelty And Significance:** 3
**Recommendation:** 5

**Clarity, Quality, Novelty And Reproducibility:**

Clarity: The paper is very clearly written and easy to understand.



Quality: The theoretical and empirical results are thorough, although I am not sure about the exhaustiveness of the comparisons (see (W2)).



Reproducibility: The results looks reproducible and a lot of experimental details are provided.

**Strength And Weaknesses:**

=== Strengths ===

(S1): The addition of the ω component to the base GNNs is clearly explained and motivated, with useful visualization showing how different values of ω induce different kinds of behaviour.

(S2): The theoretical results, showing how repeated application of the propagation operator with a given value of ω corresponds to gradient descent steps on an appropriately defined energy, are interesting.

(S3): The paper is well-written and easy to read.



=== Weaknesses ===

(W1): The paper seems to be biased towards GCN-style GNNs, and largely omits generalized message-passing-based ones. It focuses on GCN and GAT, as these are the kinds of GNNs where most of the statements made in this work apply. However, GNNs based on general message passing don't seem to be affected by the pitfalls discussed here. GCN in its plain form usually doesn't work very well anyway, due to being rather constrained (as the authors note), and often more flexible variants such as GGNN [1] or PNA [2] work better (in practice, the latter is a good default choice). Both the message function and the update function can be implemented with arbitrary MLPs, and as far as I understand, GNNs that do so don't suffer from the "non-negativity" problems that the authors seek to solve. The statement that the propagation is often shared across channels doesn't seem to apply to things like GGNN, while the statement that it is shared across layers doesn't seem to apply to most practical models (it seems un-tied weights tend to work better due to higher flexibility?). Things like PNA are available in `pytorch_geometric` (which the authors use), so a direct comparison could be straight-forward. In any case, I think the discussion could be adjusted to better position this work in relation to a larger body of papers on GNNs, noting how these other models don't suffer from the problems the authors focus on.

(W2): While the empirical results look good compared to the models included in the tables, I'm not sure to what extent are these comparisons exhaustive. For example, [3] shows better results on PROTEINS, while [4] better results on ogbn-arxiv. Did the authors intend to compare to all the available results? Claiming that e.g. ωGNNs are SotA among a particular style of GNNs is a reasonable thing to aim for too, but I'd like to understand exactly what the claim here is.



=== References ===

[1] Li et al., "Gated Graph Sequence Neural Networks"

[2] Corso et al., "Principal Neighbourhood Aggregation for Graph Nets"

[3] Zhang et al., "Hierarchical Graph Pooling with Structure Learning"

[4] Sun et al., "Adaptive Graph Diffusion Networks"



=== Nitpicks ===

Here I include some final nitpicks, which did not affect my score; they are here just to help improve the paper.

- "indistinguishable from one and other": maybe "one another"?

- missing space before parenthesis in Section 2.1

- first sentence of Section 4.2 reads off

**Summary Of The Paper:**

This paper proposes an extension to GCN-style GNNs, where the propagation operator is extended with a ω factor. This allows for expressing non-smoothing transformations. The authors show theory backing the design of their model, and then perform a sequence of experiments to show its effectiveness.

**Summary Of The Review:**

While from the theoretical side the work looks strong, I am not yet sure if it's SotA in practice when compared to all the available results, and the current discussion of the GNN landscape seems biased. Therefore, for my initial evaluation I lean towards rejection, but I'm happy to adjust it after discussions with the authors and the other reviewers.

---

> ### Author Response · Authors · 2022-11-17
> **Response**
>
> We thank the review for the insightful comments and we now address them one by one.
>
> **Regarding references (W1):**
> Thank you for noting those important references. We will include and discuss them in our revised version. We would like to highlight the main differences between our $\omega$GNNs, GGNN and PNA:
> 1. **PNA proposes to aggregate information from neighboring nodes by a set of fixed propagation operators** such as [mean, std, min, max] as well as degree-based scalars, followed by an MLP. It is important to note that all those aggregators and degree-based scalars are pre-defined while **in our $\omega$GNN formulation all the weights are learnt and are therefore more flexible**. We also would like to highlight that while PNA can yield negative values by its aggregators (e.g., if the minimal value is smaller than 0), it is dependent on the input data. In our work however, $\omega$GNNs have the freedom to create both positive/negative aggregation weights regardless of the initial feature values.
>
> 2. GGNN proposes a recurrent network which in concept is different than ours (we offer a feed-forward neural network rather than a recurrent network). More crucially, GGNN relies on some random state and iterates until converges. Therefore, the basic assumptions of GGNN are significantly different than the ones made in our large number of datasets and experiments that we conducted in our paper (for example, in all datasets in our paper we have some initial features)
>
> **Regarding experimental comparison with PNA (W1)**: We will include a direct comparison to PNA on several datasets in our revised paper, as follows:
>
> 1. We first compare the MAE metric on ZINC which was reported in PNA in the table below. It can be seen that our $\omega$GCN and $\omega$GAT offer better (lower MAE) results on this dataset.
>
> | Method      | ZINC , without edge features | ZINC, with edge features |
> |-------------|------------------------------|--------------------------|
> | PNA         | 0.320                        | 0.188                    |
> | wGCN (ours) | **0.221**                        | **0.136**                    |
> | wGAT (ours) | 0.232                        | 0.147                    |
>
>
> 2. To further validate the efficacy of our method, we ran PNA on the 10 splits from Geom-GCN (Pei et al., 2020) on Cora/Citeseer/Pubmed which we also use our experiments in Tab. 3.
> We report the results in the table below. It can be seen that both our$\omega$GCN and $\omega$GAT obtain higher accuracy than PNA.
>
> | Method      | Cora  | Citeseer | Pubmed |
> |-------------|-------|----------|--------|
> | PNA         | 87.71 | 76.03    | 88.91  |
> | wGCN (ours) | **89.30** | 77.88    | 90.45  |
> | wGAT (ours) | 89.25 | **78.01**    | **90.65**  |
>
> 3. Additionally, we now compare our $\omega$GNN with PNA on consider the ogbn-arxiv where our $\omega$GCN obtains 73.02\% accuracy while PNA achieves 72.51 \%.
>
> 4. Finally, we also experiment with PNA on the PPI dataset (as reported in Tab. 5) where our $\omega$GCN obtains a score of 99.60 while PNA achieves 99.40.
>
> We believe that those results further demonstrate the experimental / performance value of our method, in addition to its simplicity and current contributions.
>
>
> **Regarding (W2):**
>  We thank you for noting [3] and [4] and we will include them in our comparisons and revised paper.  Specifically:
>
> Our $\omega$GCN obtains 73.02\% accuracy compared to 73.41\% of [3] on ogbn-arxiv, and $\omega$GAT obtains 84.4\% accuracy on PROTEINS compared to 84.91\% of [4].
>
> Our main claim is that our $\omega$GNNs offer a minimal change to networks like GCN and GAT that solves the over-smoothing problem, enables to learn mixed-sign propagation operators (which are not available in models like GCN or GAT), and offers better or similar results to state-of-the-art methods.
>
> We would like to add that even with [3,4] which indeed obtain better results than ours (we will add them to our paper), we still show state-of-the-art accuracy on 13 other datasets, including heterophilic datasets.
>
> **Regarding nitpicks/suggestions**: We are grateful for the suggestions. We will incorporate them in our revised paper.
>
> **Regarding quality score and summary**: We thank you for all the useful and insightful comments. In our revised version we will add a discussion of the papers [1-4] to provide a more general overview of other types of GNNs and a more exhaustive comparison in our experimental section, including a direct comparison to PNA [1]. We hope that you find our response satisfactory, and we would love to hear from you.

---

### Public Comment · ~Sitao_Luan1 · 2022-11-15
**Relevant Works**

Thank the authors for proposing $\omega$GNNs and the idea of learning to mix smoothing and sharpening propagation operators is quite interesting and meaningful. I would like to highlight two relevant works: [1] proposes filterbank GNNs, which use low-pass and high-pass filters to extract the smooth and non-smooth components in graph signals and combine them adaptively; [2] proposes Adaptive Channel Mixing(ACM) framework, which includes low-pass, high-pass and identity channels and uses a node-wise channel mixing mechanism to combine the channel information adaptively. Good luck to your rebuttal.

[1] Luan S, Zhao M, Hua C, et al. Complete the missing half: Augmenting aggregation filtering with diversification for graph convolutional networks[J]. GLfrontier workshop (oral), NeurIPS 2022. arXiv:2008.08844, 2020.

[2] Luan S, Hua C, Lu Q, et al. Revisiting Heterophily For Graph Neural Networks[J]. NeurIPS 2022. arXiv:2210.07606, 2022.

---

> ### Author Response · Authors · 2022-11-17
> **Thank you for the references**
>
> We thank you for the references which we will add to our revised paper.

---

### Author Response · Authors · 2022-11-17
**General comment**

Dear reviewers,

We would like to thank you all for your detailed, constructive and insightful comments.
In our rebuttal we address all of your concerns individually, and we hope that you find it satisfactory.

With kind regards,

The authors.

---

### Decision · Program_Chairs · 2023-01-20

**Decision:**

Reject

**Justification For Why Not Higher Score:**

There is a general consensus between all reviewers that major work is needed before this work would pass the bar for ICLR.

**Justification For Why Not Lower Score:**

N/A

**Metareview: Summary, Strengths And Weaknesses:**

The authors propose a method to control oversmoothing in GNNs by introducing a weighting factor $\omega$, applied channel-wise, to mix various propagation operators at once. All reviewers agree that the context is valuable, and all of them had positive things to say about the paper (as evidenced by the reviews, praising the work's easy-to-understand motivation and solid execution). However, even after the authors' rebuttal, significant concerns remain about the soundness of the work's theory, significance of the experimental results, and the relative novelty with respect to the related work. The only possibility in the current state is to reject the paper, unfortunately. But I do encourage the authors to continue developing their work further!